# Auranofin prevents liver fibrosis by system *Xc*-mediated inhibition of NLRP3 inflammasome

Hyun Young Kim[1], Young Jae Choi[2], Sang Kyum Kim[2], Hyunsung Kim [3], Dae Won Jun [4], Kyungrok Yoon[1], Nayoun Kim[1], Jungwook Hwang [5], Young-Mi Kim[6], Sung Chul Lim[7] & Keon Wook Kang [1✉]

Demand for a cure of liver fibrosis is rising with its increasing morbidity and mortality. Therefore, it is an urgent issue to investigate its therapeutic candidates. Liver fibrosis progresses following 'multi-hit' processes involving hepatic stellate cells, macrophages, and hepatocytes. The NOD-like receptor protein 3 (NLRP3) inflammasome is emerging as a therapeutic target in liver fibrosis. Previous studies showed that the anti-rheumatic agent auranofin inhibits the NLRP3 inflammasome; thus, this study evaluates the antifibrotic effect of auranofin in vivo and explores the underlying molecular mechanism. The antifibrotic effect of auranofin is assessed in thioacetamide- and carbon tetrachloride-induced liver fibrosis models. Moreover, hepatic stellate cell (HSC), bone marrow-derived macrophage (BMDM), kupffer cell, and hepatocyte are used to examine the underlying mechanism of auranofin. Auranofin potently inhibits activation of the NLRP3 inflammasome in BMDM and kupffer cell. It also reduces the migration of HSC. The underlying molecular mechanism was inhibition of cystine-glutamate antiporter, system *Xc*. Auranofin inhibits system *Xc* activity and instantly induced oxidative burst, which mediated inhibition of the NLRP3 inflammasome in macrophages and HSCs. Therefore, to the best of our knowledge, we propose the use of auranofin as an anti-liver fibrotic agent.

[1] College of Pharmacy and Research Institute of Pharmaceutical Sciences, Seoul National University, Seoul, Republic of Korea. [2] College of Pharmacy, Chungnam National University, Daejeon, Republic of Korea. [3] Department of Pathology, Hanyang University College of Medicine, Seoul, Republic of Korea. [4] Department of Internal Medicine, Hanyang University College of Medicine, Seoul, Republic of Korea. [5] Graduate School of Biomedical Science and Engineering, Hanyang University, Seoul, Republic of Korea. [6] College of Pharmacy and Institute of Pharmaceutical Science and Technology, Hanyang University, Ansan, Gyeonggi-do, Republic of Korea. [7] College of Medicine, Chosun University, Gwangju, Republic of Korea. ✉email: kwkang@snu.ac.kr

Chronic liver injury resulting from non-alcoholic fatty liver disease or alcoholic hepatitis ultimately leads to liver fibrosis[1,2]. The demand for a cure is rising with the increased morbidity and mortality of liver fibrosis. However, the US Food and Drug Administration has not approved any drug as a first-line therapeutic option for liver fibrosis[3,4]. Therefore, therapeutic drug candidates are needed. Hepatic fibrosis progresses through a "multi-hit" process involving death of hepatocytes, cytokine secretion from macrophages, and activation of hepatic stellate cells (HSCs)[5]. Despite growing research interest, little is known regarding the potential therapeutic target(s) that control the "multi-hit" process.

Researchers have suggested that blocking the NOD-like receptor protein 3 (NLRP3) inflammasome attenuates the "multi-hit" process[6–8]. The NLRP3 inflammasome has been identified as an essential trigger for liver inflammation in patients with non-alcoholic fatty liver disease and liver fibrosis. The NLRP3 inflammasome is a multiprotein complex that recruits the ASC adapter protein and procaspase-1 in response to noxious signals (pathogen-associated molecular patterns and death-associated molecular patterns). Activation of the NLRP3 inflammasome is controlled by a two-step process[9]. First, pathogen-associated molecular patterns induce nuclear factor-κB-dependent synthesis of NLRP3 and pro-interleukin (pro-IL-)1β after activation of Toll-like receptors. Second, danger signals (e.g., extracellular ATP or uric acid crystals) drive assembly of the inflammasome, which activates caspase-1. Activated caspase-1 catalyses IL-1β processing and induces pyroptotic cell death with gasdermin cleavage. Previous studies have shown that the NLRP3 inflammasome is critical for the progression of fibrosis by means of macrophage and HSC regulation[6,8,10,11]. Notably, Mridha et al.[12] identified MCC950, a specific NLRP3 inhibitor, as a potential therapeutic candidate to attenuate severe liver fibrosis.

Auranofin has been clinically used as an anti-rheumatic agent since 1985. We have reported that auranofin suppresses the differentiation of osteoclasts by potently inhibiting the NLRP3 inflammasome[13]. Based on the notion that the NLRP3 inflammasome plays a key role in liver fibrogenesis, the potential for clinical use of auranofin to treat liver fibrosis was assessed in two experimental liver fibrosis models. Moreover, we elucidated the molecular mechanism underlying the effects of auranofin on the NLRP3 inflammasome. The Xc antiporter system is important for cellular redox homeostasis by means of cysteine uptake regulation. Here, we show that auranofin inhibited the NLRP3 inflammasome by blocking the Xc antiporter system.

## Results

**Auranofin inhibits liver fibrogenesis**. To determine whether auranofin inhibits liver fibrosis, Balb/c mice were intraperitoneally injected with thioacetamide (100 mg/kg) for 8 weeks, twice per week. Concurrently, the mice were orally administered vehicle or auranofin (1, 3, and 10 mg/kg) five times per week. Histological examinations revealed a reduction in the liver fibrotic area and fibrosis score of auranofin-administered mice (Fig. 1a, b). Liver weight relative to body weight increased by the injection of thioacetamide was alleviated by administration of auranofin (Fig. 1c). Quantification of profibrotic markers in fibrotic liver tissues demonstrated similar results. Hepatic protein expression levels of α smooth muscle actin (αSMA) decreased in a dose-dependent manner (Fig. 1d). Hepatic mRNA expression levels of Acta2, Col1a1, and Timp1 (Fig. 1e) decreased in the auranofin-administered group. Moreover, administration of auranofin significantly abrogated the increase of serum alanine aminotransferase (ALT) and aspartate aminotransferase (AST) levels in thioacetamide-injected mice (Fig. 1f).

Another common method to induce experimental liver fibrosis in mice involves periodic exposure to carbon tetrachloride $(CCl_4)$[14]. C57BL6J mice were intraperitoneally injected with 0.5 ml/kg carbon tetrachloride for 3 weeks, twice per week. Auranofin (1, 3, or 10 mg/kg) was also orally administered to these mice, five times per week. Mice injected with carbon tetrachloride showed enhanced hepatic collagen deposition, as determined by Masson's trichrome staining. Collagen deposition and collagen I protein levels in liver tissues decreased in response to auranofin. Moreover, expression of αSMA, a representative HSC activation marker, was also diminished in the auranofin-treated group (Supplementary Fig. 1). These results suggest that auranofin suppresses liver fibrosis in experimental animal models.

**Auranofin inhibits NLRP3 inflammasome-mediated IL-1β secretion in BMDM and kupffer cell**. Kupffer cells or infiltrating macrophages play a major role in the pathogenesis of chronic liver diseases[15]. Thus, we first studied the effects of auranofin on the inflammatory response of BMDM (Fig. 2a and Supplementary Fig. 2). Auranofin reduced the inflammatory responses (NOS2 and COX2 expression) induced by lipopolysaccharide (LPS) at a very low concentration (0.03 μM). The NLRP3 inflammasome has been identified as a trigger for liver fibrosis in patients with non-alcoholic fatty liver disease[7]. Hence, we hypothesized that inhibition of inflammasome activation would contribute to the anti-inflammatory effect of auranofin. Protein expression of inflammasome components (e.g. NLRP3, ASC, procaspase-1, mature IL-1β, and caspase-1 (p20)) was enhanced in thioacetamide-induced fibrotic liver tissue lysates, whereas auranofin reduced the protein level of mature IL-1β and caspase-1 (p20), markers of inflammasome activation (Fig. 2b). To elucidate the effect of auranofin on activation of the NLRP3 inflammasome in BMDM, we investigated the effect of auranofin on activation of caspase-1 and release of IL-1β. Exposure of LPS-primed BMDM to 1 mM ATP sharply increased IL-1β secretion; auranofin (0.01–0.3 μM) cotreatment with ATP suppressed IL-1β secretion in a concentration-dependent manner (Fig. 2c). Western blot analysis confirmed that auranofin (0.03 μM) potently reduced the amounts of caspase-1 (p20) and mature IL-1β in culture media of BMDM exposed to LPS and ATP (Fig. 2d). The formation of NLRP3-dependent ASC oligomers is a key event during activation of the NLRP3 inflammasome. Immunocytochemistry analysis revealed that fewer specks were assembled in BMDM treated with 0.1 μM auranofin (Supplementary Fig. 2).

Modified LDL and circulating free fatty acids (FFAs) are endogenous mediators that have been found to activate the NLRP3 inflammasome[16]. In an attempt to evaluate the effect of auranofin on NLRP3 inflammasome-activated by FFAs, LPS-primed BMDMs were incubated with palmitic acid and auranofin for 20 h. Auranofin indeed inhibited LPS/palmitic acid-induced IL-1β secretion (Fig. 2e). Furthermore, mature IL-1β secretion, which is stimulated by other NLRP3 agonists (e.g. nigericin and monosodium urate crystal) was inhibited by 0.1 μM auranofin in BMDM (Fig. 2f and Supplementary Fig. 2). In contrast, a low concentration of auranofin (0.03 μM) did not suppress the activation of NLRC4, which was induced by flagellin treatment (Supplementary Fig. 2). 0.1 μM auranofin marginally inhibited the activation of AIM2 inflammasome in poly(dA:dT) transfected BMDMs (Supplementary Fig. 2). Additionally, auranofin slightly inhibited secretion of tumor necrosis factor-α, an inflammatory cytokine for which secretion is not dependent on NLRP3 inflammasome activation (Supplementary Fig. 2). Addition of auranofin to LPS-primed kupffer cells produced similar results, albeit the magnitude of IL-1β release was less. Auranofin potently

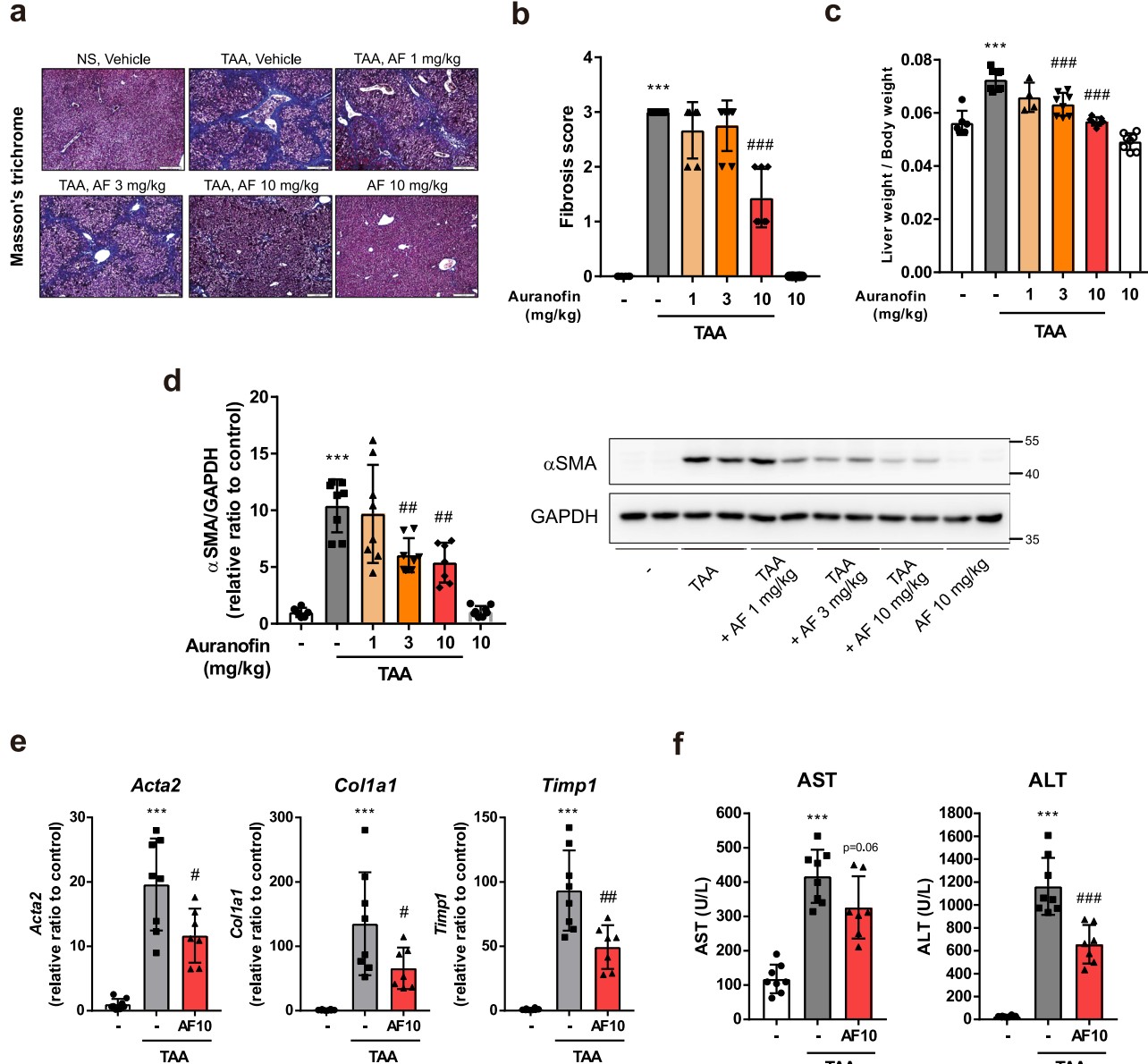

**Fig. 1 Auranofin inhibits thioacetamide-induced liver fibrosis. a** Representative Masson's Trichrome stained liver sections demonstrating collagen deposition in thioacetamide (TAA) injected mice. NS; normal saline, AF; auranofin. **b** Fibrosis score assessed by liver pathologist in a blind manner. **c** Elevation of relative liver weight was decreased by the administration of auranofin. Data are presented as mean ± SD; $n = 6$ mice for control group, $n = 7$ mice for TAA + auranofin 10 mg/kg administered group and $n = 8$ mice for the rest of the groups. **d** Quantification of αSMA by western blot analysis and **e** *Acta2*, *Col1a1*, and *Timp1* by qPCR. **f** Serum AST and ALT levels. Data are presented as mean ± SD; $n = 8$ mice for control group and TAA-administered group, $n = 7$ mice for TAA + auranofin 10 mg/kg administered group. ***$p < 0.001$, versus control group; #$p < 0.05$, ##$p < 0.01$, ###$p < 0.001$, versus TAA group by one-way ANOVA followed by Tuckey's test.

suppressed ATP- or nigericin-induced IL-1β release in LPS-primed kupffer cells (Fig. 2g).

**Auranofin inhibits pyroptosis and induces apoptosis in macrophages.** Pyroptosis is cell death characterized by membrane rupture and the release of proinflammatory intracellular contents, and spontaneous activation of the NLRP3 inflammasome results in pyroptosis[17]. Cleavage of gasdermin D (GSDMD) is an executor of pyroptosis. N-terminal fragment of GSDMD (GSDMD-N) generated by caspase-1 forms membrane pores and releases intracellular contents, including high-mobility group box 1 (HMGB1)[18]. In LPS-primed BMDM, exposure to ATP for 16 h evoked a ruptured cell morphology, whereas auranofin-cotreated BMDMs showed a shrunken morphology (Fig. 3a). The sustained

activation of the NLRP3 inflammasome by ATP induced pyroptosis, increasing GSDMD-N, caspase-1 (p20) and secretion of HMGB1 in LPS-primed BMDMs (Fig. 3b). Auranofin reduced the levels of caspase-1 (p20) and GSDMD-N, as well as the secreted form of HMGB1 (Fig. 3b). These results suggest that auranofin prevents the inflammatory death of macrophages residing in the liver and ultimately curbs the progression of liver fibrosis. However, the viability of BMDM significantly decreased in response to a relatively high concentration (1 μM) of auranofin (Fig. 3c). Western blot analysis demonstrated that auranofin enhanced the cleaved caspase-3 protein level (Fig. 3d); the apoptosis-inducing effect of auranofin in macrophages was further confirmed by flow cytometry and caspase 3/7 activity assays (Fig. 3e, f). The data show that auranofin altered the macrophage

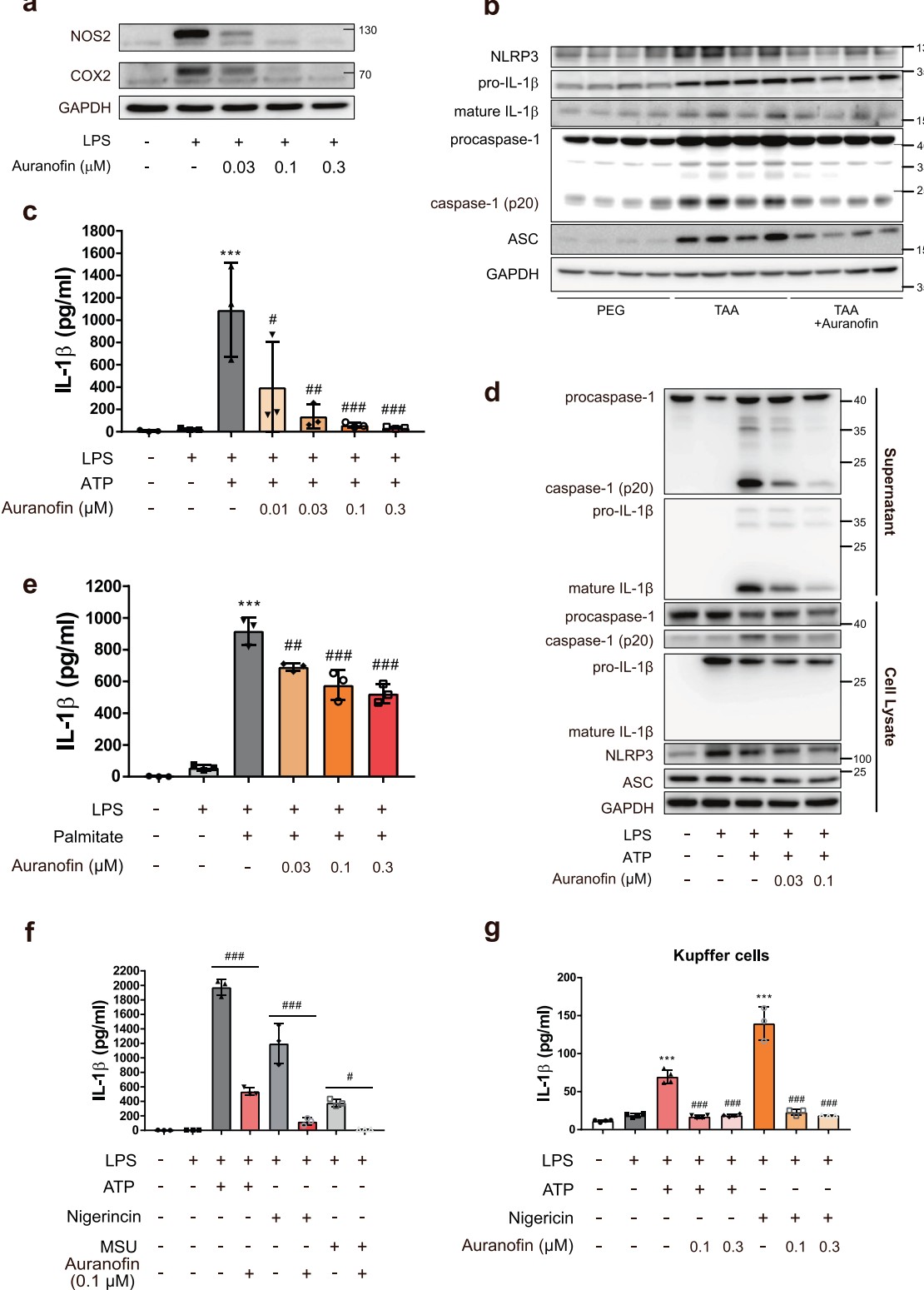

**Fig. 2 Auranofin inhibits NLRP3 inflammasome-mediated IL-1β release. a** Auranofin inhibited LPS induced inflammation in BMDM. **b** Western blot analysis of NLRP3, pro-IL-1β, mature IL-1β, procaspase-1 and ASC in liver lysates. **c**, **d** Effects of auranofin on ATP triggered IL-1β release in LPS-primed bone marrow-derived macrophage (BMDM), $n = 3$. BMDMs were primed with LPS (100 ng/ml) for 4 h, followed by exposure to ATP (1 mM) and auranofin for 1 h. **e** Effect of auranofin on palmitate (400 μM) triggered IL-1β release in LPS-primed BMDM, $n = 3$. **f** Effects of auranofin on nigericin (2 μM) or 200 μg/ml monosodium urate (MSU) triggered IL-1β release in LPS-primed BMDM, $n = 3$. **h** Effects of auranofin on ATP (1 mM) or nigericin (2 μM) triggered IL-1β release in kupffer cells, $n = 3$ or 4. Data are presented as mean ± SD (**c**, **e**, **f**, and **g**), analyzed by one-way ANOVA followed by Tuckey's test; ***$p < 0.001$, compared to control; #$p < 0.05$, ##$p < 0.01$, and ###$p < 0.001$ compared to NLRP3 inflammasome-induced group.

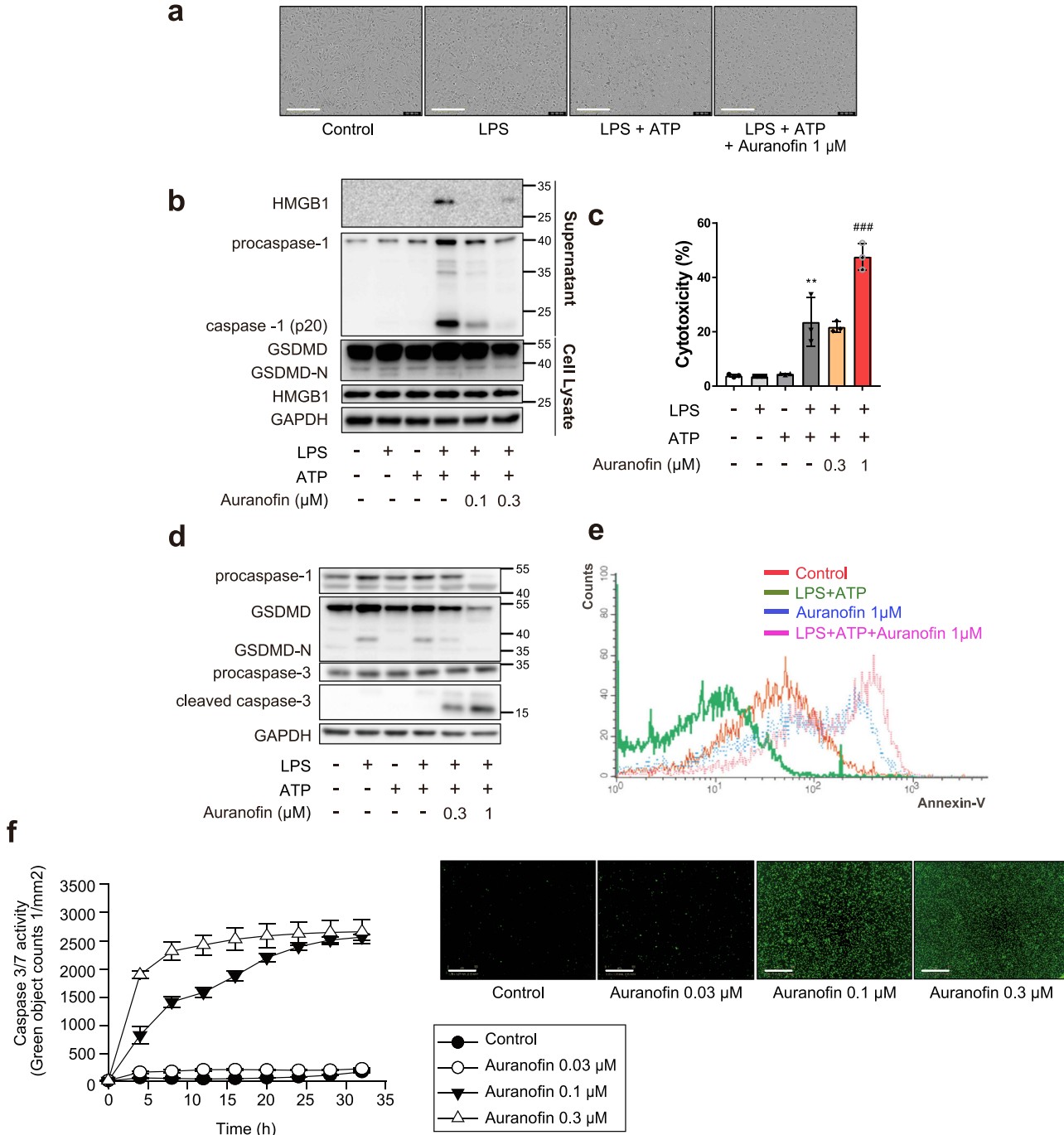

**Fig. 3 Auranofin inhibits pyroptosis and induces apoptosis in macrophages.** BMDMs were primed with LPS (100 ng/ml) for 4 h, followed by exposure to ATP (1 mM) and auranofin for 16 h. **a** Representative picture of LPS-primed BMDM treated with ATP (1 mM) and auranofin (1 μM). Scale bar = 200 μm. **b** Secretion of HMGB1 was determined by immunoblot analysis of supernatants. GSDMD-N was measured in cell lysates. **c** LDH assay was performed to evaluate cytotoxicity of auranofin on inflammasome-activated BMDM, $n = 3$. Data are presented as mean ± SD, analyzed by one-way ANOVA followed by Tuckey's test; **$p < 0.01$, compared to control; ###$p < 0.001$, compared to LPS, ATP-treated group. **d** Cleavage of caspase-3 and GSDMD-N were measured in cell lysates of BMDM primed with LPS followed by exposure of ATP (1 mM) and auranofin. **e** Annexin-V staining in the auranofin-treated BMDM. **f** Auranofin-treated BMDMs were stained using Incucyte® caspase3/7 green reagent and detected using Incucyte® ZOOM, $n = 5$. Data are presented as mean ± SEM. Scale bar = 300 μm.

death pathway from caspase-1-mediated pyroptosis to caspase-3-mediated apoptosis.

**Auranofin inhibits migration of HSCs.** HSCs are another key player in the progression of liver fibrosis. In chronic liver disease, HSCs are activated and acquire the collagen-producing myofibroblast phenotype, which contributes to liver fibrogenesis[19].

HSCs in liver tissues from thioacetamide-treated fibrotic mice could be stained with αSMA (activated HSCs) and desmin (total HSCs). Notably, the number of HSCs in the fibrotic liver was diminished in the auranofin-treated group (Fig. 4a). Hence, we investigated the effects of auranofin on activation and migration of primary mouse HSC. Quiescent HSC isolated from the liver became activated after they had been cultured with serum on an

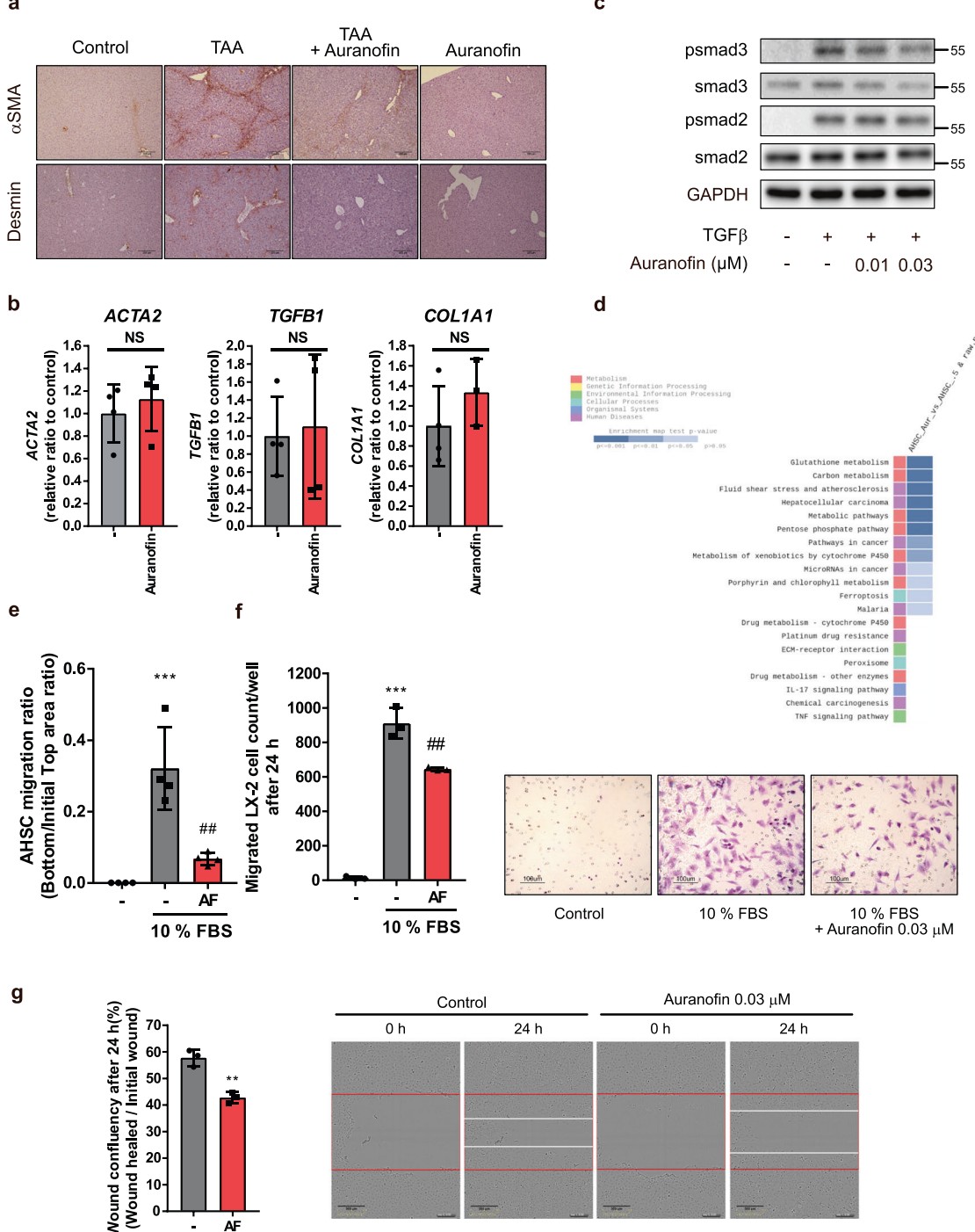

**Fig. 4 Auranofin inhibits migratory ability of activated HSCs. a** αSMA and desmin stained liver sections showing distribution of hepatic stellate cells in TAA injected mice. Scale bar = 200 μm. **b** Isolated primary mouse HSCs were cultured on uncoated plastic plates in the presence of serum and auranofin (0.03 μM) containing media. HSCs activation markers including *ACTA2*, *TGFB1*, and *COL1A1* were evaluated after incubation for 7 day, n = 4. NS; not significant. **c** Smad2/3 phosphorylation following treatment of TGFβ1 (5 ng/ml) and auranofin for 1 h. **d** KEGG pathways analysis based on gene enrichment database. RNA samples for RNA sequencing were isolated from primary HSCs exposed to vehicle or 0.03 μM auranofin for 12 h were used. **e** Migration of HSCs was measured by Incucyte® chemotaxis assay, n = 4. **f** Migration of LX-2 measured by transwell assay. Migrated LX-2 cells were stained with 0.1% crystal violet and counted per well, n = 3. Data are presented as mean ± SD (**c**, **e**, and **f**), analyzed by one-way ANOVA followed by Tuckey's test; ***$p <$ 0.001, compared to control; ##$p < 0.01$ compared to chemoattractant (10% FBS) added group. Scale bar = 100 μm. AF; auranofin 0.03 μM. **g** Effect of auranofin on wound healing of LX-2 cells. Scratch closure changes were measured after creating wounds on cells using 96-well ImageLock plate and incubation with auranofin-containing media, n = 3. Scale bar = 300 μm. Data are presented as mean ± SD, analyzed by unpaired student's t-test. AF; auranofin 0.03 μM, **$p < 0.01$ compared to healed wound confluency of control.

uncoated plastic plate; this activation was demonstrated by the enhancement of αSMA expression (Supplementary Fig. 3). However, primary HSCs cultured in media containing 0.03 μM auranofin for 7 days did not show any changes in mRNA levels of profibrotic markers (Fig. 4b). Moreover, auranofin did not reduce transforming growth factor-β-induced phosphorylation of smad2/3, which stimulates transcription of profibrotic genes (Fig. 4c).

Extracellular matrix receptor interaction was a key feature differentially altered in auranofin-treated HSCs, as determined by KEGG analysis of RNA sequencing results in primary HSCs exposed to vehicle or 0.03 μM auranofin (Fig. 4d). As a cell–matrix interaction is necessary to anchor HSCs, we checked the effect of auranofin on the migratory ability of HSCs. HSCs acquire migratory ability to spread throughout the perisinusoidal space and then perpetuate to the fibrotic stage. Auranofin potently inhibited the migratory ability of activated HSC (Fig. 4e). Consistent with this finding, the inhibitory effect of auranofin on HSC migration was confirmed in the LX-2 human stellate cell line by IncuCyte® chemotaxis, transwell, and wound healing assays (Fig. 4f, g and Supplementary Fig. 3). These data indicate that auranofin prevents liver fibrosis by inhibiting the migration of HSCs.

It has been recently reported that secreted HMGB1 is involved in HSC migration[20]. We assumed that a reduction of HMGB1 secretion, following inhibition of the NLRP3 inflammasome by auranofin, would hinder HSC migration. Similar to macrophages, NLRP3 inflammasome activity triggered by LPS and ATP was significantly alleviated by auranofin in activated HSCs (Supplementary Fig. 3). These data suggest that the inhibitory effect of auranofin on the NLRP3 inflammasome contributes to reduced HSC migration.

**Auranofin depletes glutathione (GSH) and induces oxidative burst, a key factor in the inhibition of IL-1β secretion by macrophages.** We investigated the potential mechanism by which auranofin suppresses NLRP3 activation in macrophages. When we assessed the differentially expressed genes in BMDM exposed to vehicle or auranofin for 1 h, we detected glutathione peroxidase 1 (Gpx1), a representative GSH-related antioxidant gene (Fig. 5a). Gpx1 is an enzyme that protects cells against oxidative stress by catalyzing GSH-dependent reduction; thus, we determined intracellular GSH levels in BMDM. Unexpectedly, exposure to auranofin for 1 h depleted GSH in BMDM (Fig. 5b).

To assess if the depletion of GSH in auranofin-treated BMDM is related to its anti-inflammasome effect, we quantified intracellular GSH and IL-1β secretion levels in BMDMs treated with GSH-depleting reagents. Buthionine sulfoximine (BSO), a specific inhibitor of γ-glutamylcysteine-ligase and diethylmaleate (DEM), a GSH conjugating reagent depleted intracellular GSH within 5 and 1 h, respectively (Supplementary Fig. 4). Treatment of BSO or DEM significantly decreased the inflammasome-mediated IL-1β secretion in LPS/ATP-treated BMDMs (Fig. 5c, d).

Concurrently, we observed that the intracellular reactive oxygen species (ROS) level increased in response to auranofin in a concentration-dependent manner (Fig. 5e). Although the relationship between ROS production and inflammasome activation remains controversial, transient ROS production caused by tert-butyl hydroperoxide significantly inhibited inflammasome activation (Fig. 5f). We further examined whether auranofin-mediated suppression of inflammasome activation was due to the generation of ROS. Cotreatment with N-acetyl cysteine and auranofin reversed the inhibitory effects of auranofin on both inflammasome activation (mature IL-1β and caspase-1 (p20))

(Fig. 5g and Supplementary Fig. 4) and pyroptosis induction (GSDMD-N) (Fig. 5h). Additionally, cotreatment of BMDMs with auranofin and N-acetyl cysteine did not induce caspase-3/7-dependent apoptosis (Fig. 5i, j). Hence, GSH depletion and the subsequent intracellular ROS generation by auranofin may contribute to the inhibitory effects on NLRP3 inflammasome.

**Reduced hepatocyte sensitivity to auranofin is related to enhancement of GSH levels.** Hepatocytes are relatively resistant to auranofin cytotoxicity. The IC50 value of auranofin-mediated cell death was much higher in primary hepatocytes than in BMDMs and HSCs (Fig. 6a); this finding suggested that hepatocytes are resistant to auranofin activity at a concentration that caused antifibrotic effects in macrophages and HSCs. As inhibition of the inflammasome by auranofin was mediated through GSH depletion in BMDM, we assessed intracellular GSH levels in auranofin-treated primary hepatocytes. The basal level of total GSH was approximately tenfold greater in primary hepatocytes than in BMDMs and HSCs (Fig. 6b). Whereas, the intracellular GSH level in primary hepatocytes did not significantly change after 1 h of exposure to 0.3 μM auranofin (Fig. 6c), suggesting that the abundant GSH pool protects primary hepatocytes from oxidative burst and auranofin treatment. Also, the components of the NLRP3 inflammasome (e.g., NLRP3 and ASC) were deficient in primary hepatocytes (Fig. 6d). Hence, the hepatocytes could not secrete a significant quantity of IL-1β under the activated NLRP3 inflammasome condition (Fig. 6e). These data suggest that auranofin specifically targets macrophages and HSC, which contain components to assemble the NLRP3 inflammasome and have relatively low levels of the GSH pool.

**Auranofin inhibits system *Xc* in macrophages and activated HSCs.** Auranofin has been reported to inhibit thioredoxin reductase[21]. However, we found that BMDM and HSC lack thioredoxin reductase (Fig. 7a). Moreover, unlike auranofin, D9 (thioredoxin reductase inhibitor) did not inhibit NLRP3-dependent IL-1β secretion (Fig. 7b). These results suggest that thioredoxin reductase is not a primary target for the inhibitory effect of auranofin on the NLRP3 inflammasome. Auranofin is an organometallic compound containing gold (Au); thus, inductively coupled plasma-mass spectrometry was used to detect the ratio of auranofin distributed in cell lysates to culture media. Notably, only 10–20% of the exposed auranofin was detected in cell lysates from BMDMs at 1 h after auranofin treatment (Fig. 7c), demonstrating that most auranofin did not penetrate the plasma membrane.

The system *Xc* cystine/glutamate antiporter regulates oxidative stress by importing cystine, the oxidized form of cysteine[22]. As cysteine is a major building block for GSH, we hypothesized that the inhibiting system *Xc* may serve as a basis for the instant oxidative bursts following auranofin treatment. System *Xc* is composed of a light-chain subunit, xCT (*SLC7A11*), and a heavy chain subunit 4F2hc (*SLC3A2*). System *Xc* activity is modulated by the expression of xCT in the plasma membrane, which is stabilized by interaction with the cell surface marker CD44, thereby promoting GSH synthesis[23,24].

First, we identified xCT and CD44 in mouse primary cells, including activated HSC, BMDM, and hepatocyte. Whereas hepatocytes lack CD44, both xCT and CD44 were observed in BMDM and activated HSC, which exhibited potent antifibrotic effects following auranofin treatment (Fig. 7d and Supplementary Fig. 5). To investigate whether auranofin inhibits system *Xc* activity, we determined the quantity of cellular labeled cysteine (M + 3) originating from system *Xc*-mediated cystine uptake by using stable isotope-labeled cystine (Fig. 7e). The labeled cysteine

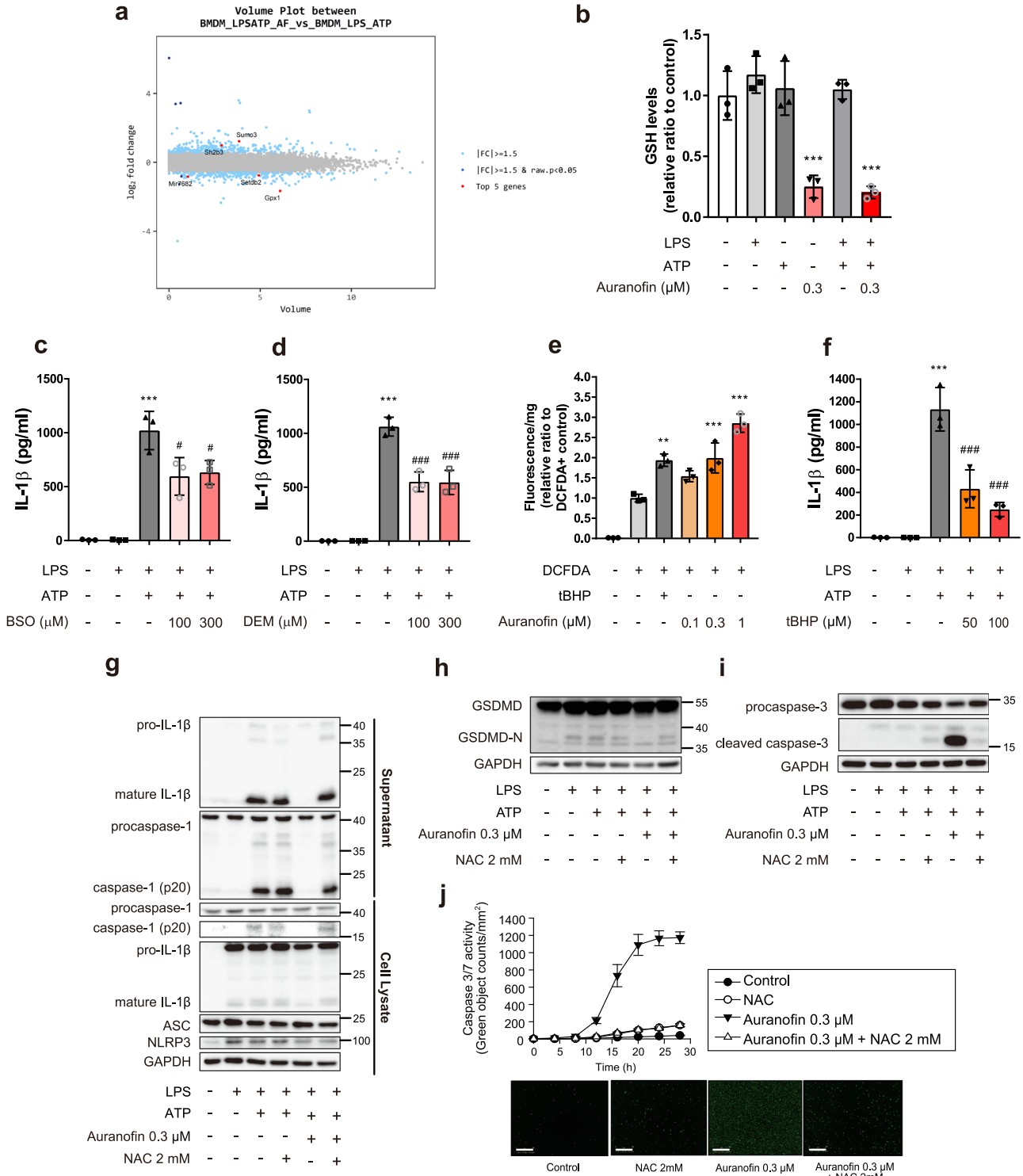

in BMDMs significantly decreased 15 min after auranofin exposure (Fig. 7f). Vice versa, the quantity of glutamate exported to the supernatant increased in response to auranofin treatment (Fig. 7g).

**Inhibiting system *Xc* leads to blockade of the NLRP3 inflammasome in profibrotic liver.** Although system *Xc* is detected at very low levels in normal human liver tissues, the level of system *Xc* expression was highly upregulated in liver tissues from patients with non-alcoholic steatohepatitis who exhibited liver fibrosis (Fig. 8a, b). In addition, most system *Xc* expression was present along the fibrotic area where αSMA expression levels were high (Fig. 8c). Light-chain subunit of system *Xc, Slc7a11* (xCT) was highly expressed in TAA-induced fibrotic liver (Fig. 8d); its expression also included the F4/80-positive area where macrophages are infiltrated (Fig. 8e). These data indicate that system *Xc* is upregulated in the profibrotic condition, particularly in infiltrated nonparenchymal cells including HSC and macrophage.

We then tested the effect of system *Xc* inhibition on NLRP3 inflammasome activation. Sulfasalazine and erastin are specific inhibitors of system *Xc*. Treatment with sulfasalazine or erastin significantly inhibited activation of the NLRP3 inflammasome in

**Fig. 5 Inflammasome inhibition effects of auranofin were mediated by ROS bursts. a** Volume plot between LPS/ATP-treated BMDM and LPS/ATP-treated BMDM with 0.3 μM auranofin. **b** Intracellular GSH levels in BMDM treated with LPS (100 ng/ml), ATP (1 mM) and auranofin (0.3 μM), $n = 3$; ***$p < 0.001$ compared to control. **c, d** Effects of GSH depleting agents on ATP triggered IL-1β release in LPS-primed BMDM. **c** BMDMs were treated with BSO (100 μM) and LPS (100 ng/ml) for 4 h, followed by exposure to ATP (1 mM) for 1 h, $n = 3$. **d** BMDMs were primed with LPS for 4 h, followed by exposure to ATP and DEM (300 μM) for 1 h, $n = 3$. **e** Intracellular ROS levels were measured by green fluorescence of DCFDA-positive BMDM, $n = 3$; **$p < 0.01$ and ***$p < 0.001$, compared to DCFDA-positive group. **f** Effects of tBHP on ATP triggered IL-1β release of LPS-primed BMDM, $n = 3$; Data are presented as mean ± SD (**c, d**, and **f**), analyzed by one-way ANOVA followed by Tuckey's test. ***$p < 0.001$ compared to control; #$p < 0.05$ and ###$p < 0.001$, compared to NLRP3 inflammasome-induced group. **g** ATP triggered IL-1β release by NLRP3 inflammasome was determined by western blot analysis in LPS-primed BMDM after cotreatment of NAC with auranofin. **h, i** BMDMs were primed with LPS (100 ng/ml) for 4 h, followed by exposure to ATP (1 mM), NAC (2 mM) and auranofin for 16 h. GSDMD-N and cleaved caspase-3 were measured using western blot analysis. **j** BMDM treated with 0.3 μM auranofin and 2 mM NAC. Apoptotic cells were counted using green fluorescence of Incucyte® caspase3/7 reagent. Data are presented as mean ± SEM, $n = 4$. Scale bar = 300 μm.

BMDM and kupffer cell (Fig. 8f–i). Furthermore, NLRP3-dependent IL-1β secretion was diminished in the cystine-deprived condition (Fig. 8j). These data confirm that system *Xc* inhibition leads to blockade of the NLRP3 inflammasome; this may be the molecular mechanism underlying the inhibitory effect of auranofin on inflammasome activation.

## Discussion

No effective therapy is available for liver fibrosis, despite the growing demand for a cure. Liver fibrosis is characterized by a multi-hit process involving a complex interplay between hepatocytes and nonparenchymal cells, including macrophages and HSCs[25]. Although there have been many studies regarding liver fibrogenesis, there is little understanding of therapeutic candidates to treat liver fibrosis. This is partly due to the lack of drug candidates that hinder fibrogenesis in a multi-targeted mechanism.

There has been growing interest in the NLRP3 inflammasome, an intracellular multi-molecular complex that regulates inflammatory signals and HSC migration[8,26]. Mice deficient in components of the NLRP3 inflammasome (NLRP3-/- and caspase-1-/- mice) are relatively insensitive to liver fibrotic stimuli[10,27]. A small molecule inhibitor of the NLRP3 inflammasome (MCC950) shows potential as a drug for liver fibrosis. However, its efficacy and safety have not yet been established.

Auranofin is a therapeutic agent clinically approved for treatment of rheumatoid arthritis; it has been recently investigated for potential therapeutic applications in a wide range of diseases[28,29]. To the best of our knowledge, this is the first in vivo study to demonstrate the feasibility of using auranofin as an antifibrotic agent. Administration of auranofin to a thioacetamide- or carbon tetrachloride-induced fibrotic liver significantly inhibited the progression of fibrosis. The underlying mechanism comprised blockade of the NLRP3 inflammasome, which resulted in reduction of IL-1β secretion and induction of pyroptosis in macrophages, as well as reduction of HSC migratory ability. It has been very recently reported that high concentration (>1 μM) gold compounds inhibit TGFβ1 signaling in LX-2 cells[30]. However, we failed to show any significant effect of auranofin (<0.1 μM) on TGFβ1-stimulated Smad2/3 phosphorylation in primary HSCs. Considering IC$_{50}$ value of auranofin on HSC cytotoxicity, submicromolar ranges of auranofin would be physiologically relevant concentration in fibrotic liver.

We propose that the inhibitory effect of auranofin on the NLRP3 inflammasome originated from the oxidative burst in macrophage and HSC. Using RNA sequencing data, we confirmed that auranofin reduced the expression levels of mRNA related to GSH metabolism. Auranofin depleted intracellular GSH and enhanced levels of ROS, resulting in NLRP3 inflammasome inhibition. The inhibitory effect of auranofin on the NLRP3 inflammasome was almost completely reversed by incubation of

cells with the cell-permeable ROS scavenger N-acetyl cysteine. Although ROS are implicated in the regulation of the inflammasome, the distinct cellular targets of ROS result in different responses; many aspects of the underlying mechanisms remain unclear[31,32]. Moreover, one study reported that moderate ROS levels act as a trigger for the NLRP3 inflammasome, while excessive ROS levels in a local area lead to inhibition of NLRP3 activation[33]. Our data indicate that a short-term ROS burst blocks the NLRP3 inflammasome signals.

The mechanism by which ROS modulates NLRP3 activity remains elusive. However, a plausible explanation for the inhibitory effect of ROS bursts on NLRP3 activation is that superoxide directly decreased caspase-1 activity. In SOD1-deficient macrophages, higher superoxide levels decreased the cellular redox potential. Consequently, it inhibited caspase-1 by reversible oxidation and glutathionylation of the redox-sensitive cysteine residues[34]. The effect of superoxide was independent of LPS priming, only inhibiting the cleavage of caspase-1. In the auranofin-treated BMDM, we confirmed the increased ROS levels using fluorescent ROS indicator CM-H2DCFDA. Based on the previous report, the intracellular ROS generated by auranofin is expected to oxidize or glutathionylate the active cysteine site of caspase-1. Indeed, auranofin did not reduce priming signals of the NLRP3 inflammasome. It only inhibited the cleavage of caspase-1 and maturation of IL-1β.

Previous studies have reported enhanced levels of intracellular ROS and reduced release of IL-1β by auranofin due to regulation by thioredoxin reductase[35,36]. However, we found that the oxidative burst was caused by the inhibitory effect of auranofin on system *Xc* activity. System *Xc* is a cystine/glutamate transporter that balances intracellular redox homeostasis. Inhibition of system *Xc* results in a lack of intracellular cysteine, which rapidly leads to oxidative stress[37]. Our study revealed that auranofin reduced the quantity of intracellular cysteine that originated from cystine imported by system *Xc*; the depletion of cysteine led to rapid reduction of GSH and enhancement of intracellular ROS in macrophages.

Pharmacological inhibition of system *Xc* attenuates alcoholic steatosis by suppressing 2-arachidonoylglycerol production in HSCs and subsequent de novo lipogenesis in hepatocytes[38]. Chronic alcohol consumption impairs the transsulfuration pathway and increases system *Xc* expression in hepatocytes as a compensatory mechanism. However, macrophages and HSCs have limited expression of transsulfuration pathway enzymes (Supplementary Fig. 5) and are expected to show a high dependency on system *Xc*, compared to hepatocytes[39]. Therefore, it is important to clarify the role of system *Xc* in macrophages and HSCs and investigate its contribution to liver fibrosis when macrophages infiltrate and HSCs proliferate. In this study, we showed that the expression of system *Xc* was significantly enhanced in liver samples from patients with non-alcoholic

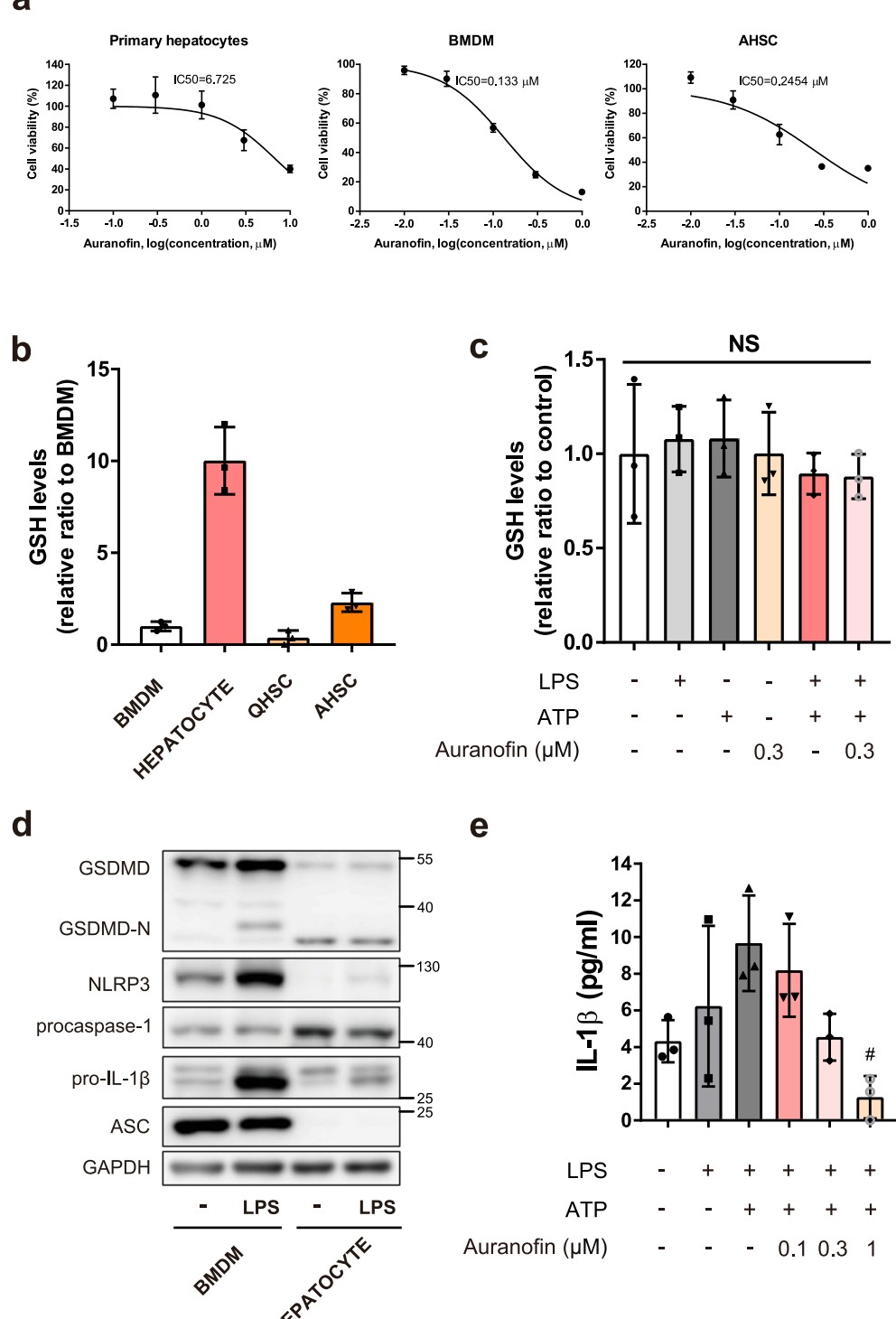

**Fig. 6 Hepatocytes are resistant to the effects of auranofin. a** IC50 values of auranofin were determined using MTT assay on mouse primary cells including hepatocyte, BMDM, and AHSC. AHSC; Activated hepatic stellate cell. $n = 7$, $n = 5$, and $n = 4$ per group for hepatocyte, BMDM and AHSC, respectively. **b** Intracellular GSH levels of hepatocyte, BMDM and HSCs, $n = 3$. QHSC; Quiescent hepatic stellate cell. **c** Intracellular GSH levels of primary hepatocytes followed by LPS (100 ng/ml), ATP (1 mM) and auranofin (0.3 µM) treatment, $n = 3$. Data are presented as mean ± SD, analyzed by one-way ANOVA followed by Tuckey's test. NS; not significant. **d** Protein levels of GSDMD, GSDMD-N, proIL-1β, NLRP3, procaspase-1, and ASC were evaluated by western blot analysis. **e** Primary hepatocytes were primed with LPS (100 ng/ml) for 4 h, followed by exposure to ATP (1 mM) for 1 h. IL-1β release by NLRP3 inflammasome activation was measured by ELISA. Data are presented as mean ± SD, analyzed by one-way ANOVA followed by Tuckey's test. $n = 3$, #$p < 0.05$, compared to NLRP3 inflammasome-induced group.

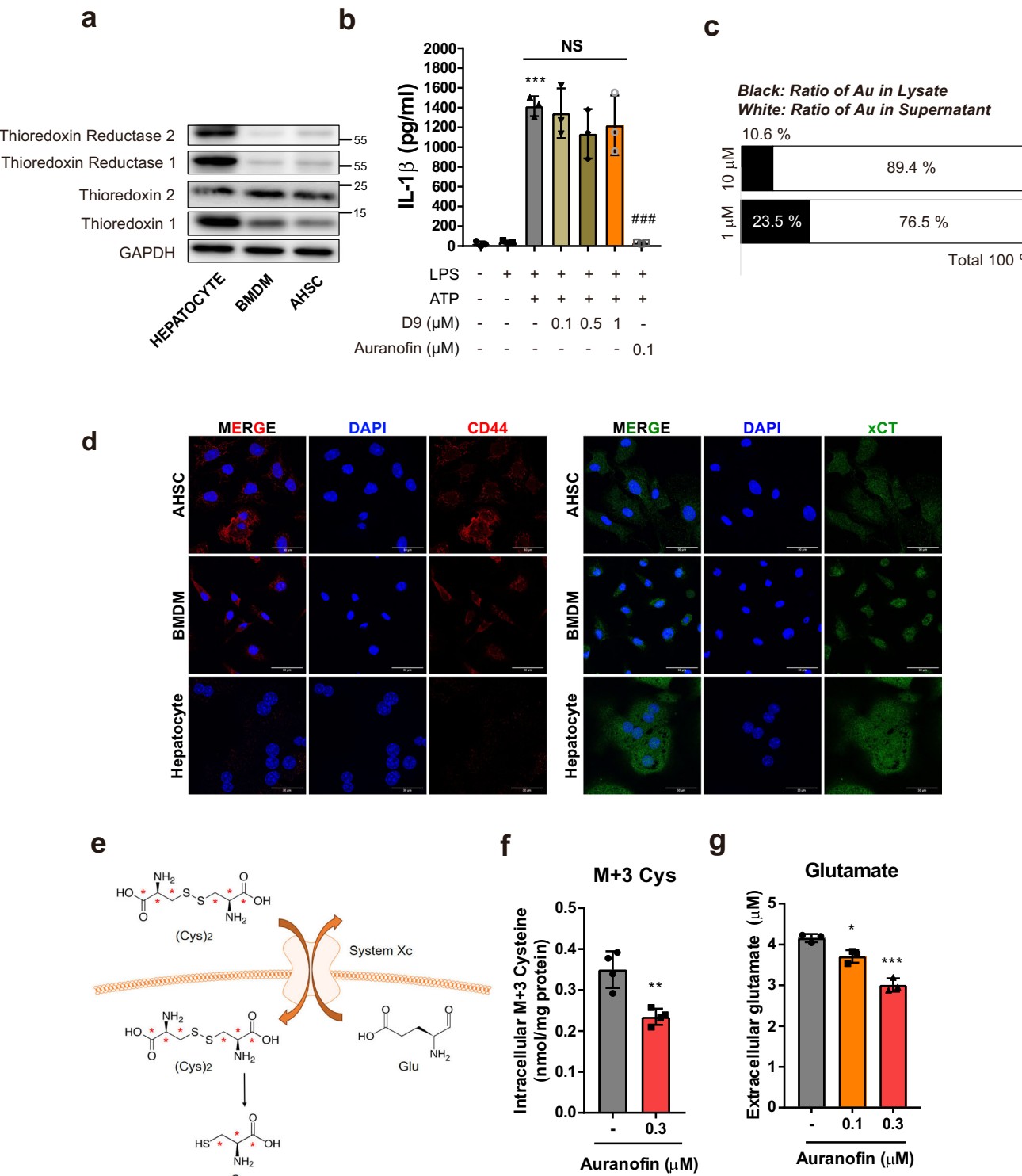

**Fig. 7 The primary target of auranofin is system _Xc_ in macrophages and activated HSCs. a** Intracellular protein levels of thioredoxins and thioredoxin reductases in hepatocyte, BMDM and activated HSC. **b** Effects of thioredoxin reductase inhibitor, D9 on ATP triggered IL-1β release of LPS-primed BMDM, $n = 3$. Data are presented as mean ± SD, analyzed by one-way ANOVA followed by Tuckey's test; ***$p < 0.001$, compared to control; ###$p < 0.001$, NS; not significant compared to NLRP3 inflammasome-induced group. **c** BMDMs were treated with 1 or 10 μM auranofin for 1 h. Ratio of gold elements in supernatant and cell lysates of BMDM were evaluated using ICP-MS. **d** Detection of xCT and CD44 in hepatocyte, BMDM and AHSC by immunofluorescence staining. Scale bar = 30 μm. **e, f** Using stable isotope of cystine (M + 6), intracellular M + 3 cysteine were measured in BMDM after treatment with auranofin (0.3 μM) for 15 min, $n = 4$. Cys; cysteine. **g** After exposure of BMDM with auranofin for 3 h, extracellular glutamate levels were measured by glutamate assay kit in supernatant of BMDM, $n = 3$. Data are presented as mean ± SD (**f** and **g**), analyzed by unpaired student $t$-test; *$p < 0.05$, **<$p < 0.01$ and, ***$p < 0.001$ compared to control.

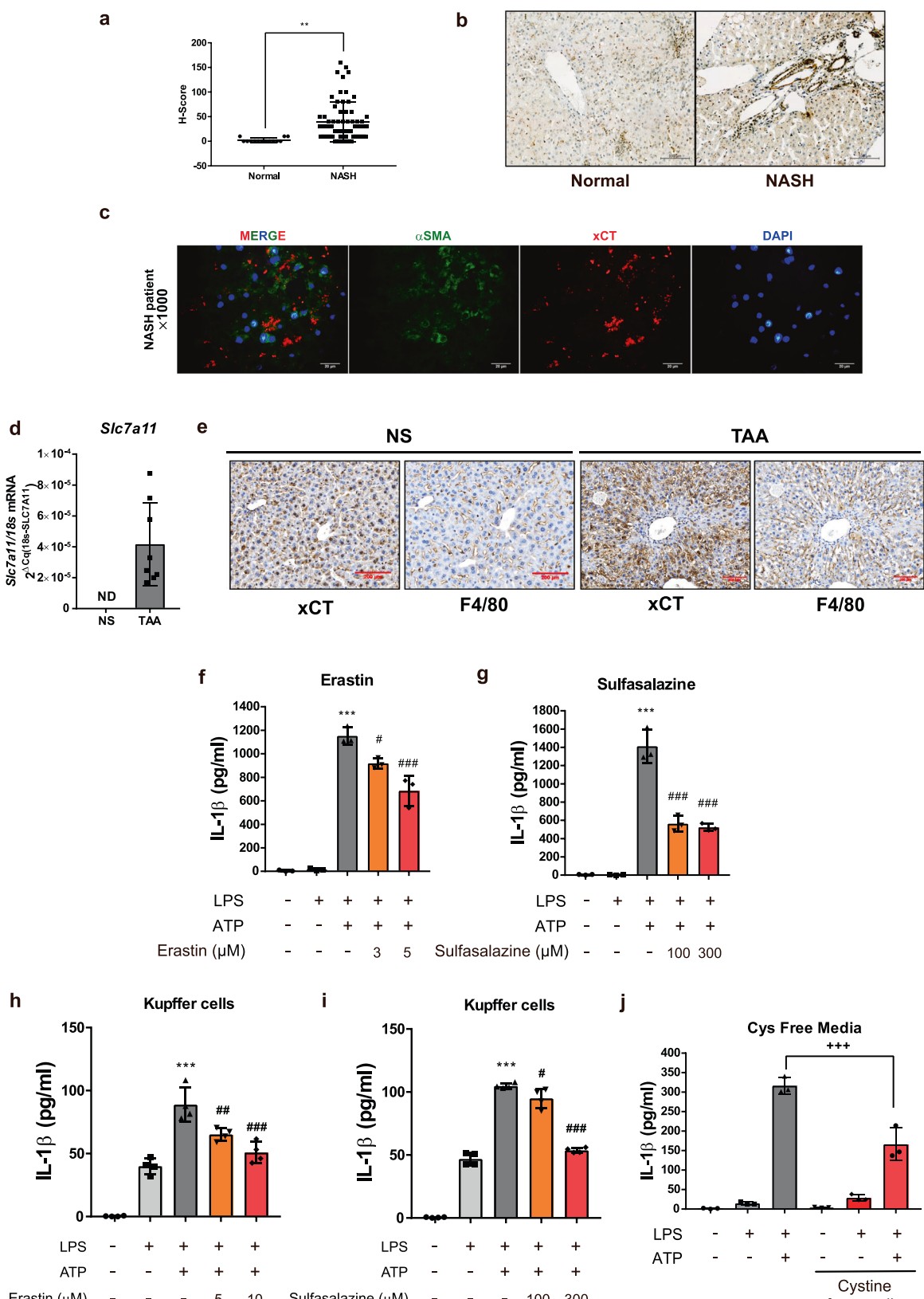

steatohepatitis whose pathological stage preceded liver fibrosis. The enhanced level of system $Xc$ activity in liver tissues was mainly localized in macrophages and HSCs. These data support the notion that the anti-liver fibrotic effect of auranofin may result from system $Xc$-mediated inhibition of the NLRP3 inflammasome in macrophages and HSCs. When we determined the protein level of pro-IL-1β in BMDMs, LPS-inducible pro-IL-1β expression was not altered by 1 h incubation with NAC or xCT inhibitors (Supplementary Fig. 4), which implies that xCT-mediated redox change is not related to LPS priming signal.

Based on the novel mechanism of action of auranofin, our findings further indicate that inhibition of system $Xc$ attenuates

**Fig. 8 System *Xc* is upregulated in profibrotic conditions and its blockade prevents activation of NLRP3 inflammasome. a** H-score based on xCT stained IHC of liver samples from 14 normal and 64 NASH patients. Data are presented as mean ± SD, analyzed by unpaired student *t*-test; **$p < 0.01$. **b** Representative images of xCT stained IHC in normal and NASH liver. Scale bar = 100 μm. **c** Immunofluorescence images showing the localization of xCT in frozen liver sample from profibrotic NASH patient. Liver tissues were labeled with αSMA (green), xCT (red) and DAPI (blue). Scale bar = 20 μm. **d** *Slc7a11* (xCT) mRNA expression level was measured using real-time qPCR in TAA-induced fibrotic liver (*n* = 8 mice per group). NS; normal saline, ND; not detected, *Slc7a11* Cq value >35. **e** IHC images showing the localization of xCT in liver samples. Liver tissues were labeled with F4/80 and xCT. NS; normal saline, Scale bar = 200 μm. **f, g**, and **j** Effects of erastin, sulfasalazine and cystine free media on ATP triggered IL-1β release of LPS-primed BMDM. **h, i** Effects of erastin and sulfasalazine on ATP triggered IL-1β release of LPS-primed kupffer cells. Data are presented as mean ± SD (**f, g, h, i**, and **j**), analyzed by one-way ANOVA followed by Tuckey's test. *n* = 3 for **f, g, j** and *n* = 4 for **h, i**. ***$p < 0.001$, compared to control; #$p < 0.05$, ###$p < 0.001$ compared to LPS, and ATP-treated group; +++$p < 0.001$, compared to LPS, and ATP-treated group in DMEM media.

liver fibrosis. System *Xc* inhibitors, including sulfasalazine and erastin, blocked activation of the NLRP3 inflammasome, suggesting that these inhibitors have the potential to cure liver fibrosis. Therefore, a therapeutic agent for fibrotic liver could be developed by thoroughly investigating substances that inhibit system *Xc*. It is important to repurpose auranofin for treatment of liver fibrosis, as there are no available first-line therapies. The findings in this study illustrate how auranofin hinders the progression of liver fibrosis and demonstrate its treatment efficacy. The clinical efficacy of auranofin in treatment of liver fibrosis is currently being tested in a phase II clinical trial. Finally, our findings suggest that inhibition of system *Xc* could improve the progression of liver fibrosis.

## Methods
**Animal experiments**. Balb/c mice were obtained from the SLC Inc (Kotoh-cho, Japan). In order to induce liver fibrosis, Balb/c mice were intraperitoneally injected with thioacetamide (100 mg/kg) for 8 weeks, twice a week. In another liver fibrosis model, male 8-weeks-old C57BL/6 mice were injected with tetrachloride (0.5 ml/kg) for 3 weeks twice a week. The mice were acclimatized for at least 1 week prior to use, and were housed in groups of 4 or 5 with a 12 h light/dark cycle at a controlled temperature of $21 \pm 2\,°C$ and $50 \pm 5\%$ humidity. The animal handlings and experimental procedures were approved by IACUC (Institutional Animal Care and Use Committee, SNU-171127-2, SNU-181105-6-1) in Seoul National University.

**Assessment of liver injury in fibrotic liver**. Serum aspartate transaminase (AST) and alanine transferase (ALT) were measured using Spectrum® (Abbott Laboratories, Abbott Park, IL, USA). Liver samples were fixed in 10% neutral-buffered formalin solution (Sigma, St. Louis, MO, USA), embedded in paraffin, and stained for histology. H&E and Masson's trichrome stained liver sections were assessed by liver pathologist in a blind manner.

**Isolation and culture of bone marrow-derived macrophages**. Bone marrow cells were isolated from 8-week-old male C57BL/6 mice by flushing the femur and tibia of with culture media. Using macrophage colony-stimulating factor (M-CSF, Peprotech Ltd, London, UK), bone marrow cells were differentiated (7 days) into bone marrow-derived macrophages (BMDMs). Bone marrow cells were incubated in RPMI medium containing 10% fetal bovine serum (FBS), 1% penicillin/streptomycin, 25 mM HEPES and 30 ng/ml M-CSF followed by media change every 3 days.

**Isolation of primary hepatocytes, kupffer cells, and HSCs**. Non-recirculating 2-step perfusion method with calcium and magnesium-free Hanks' salt solution followed by a medium containing collagenase[40] was used for the isolation of primary cells from liver tissues of C57BL/6 mice. The cell suspension was filtered through a 70 μm cell strainer and hepatocytes were obtained from the single cell-suspensions after $50 \times g$ centrifugation. Hepatocytes with viability above 90% were plated on a collagen-coated plate (Corning, New York, USA).

The supernatant was centrifuged at $600 \times g$ to obtain hepatic nonparenchymal cells. Primary HSCs were isolated by density-gradient centrifugation using ficoll paque-plus (GE Healthcare, Madison, WI) and percoll (GE Healthcare, Madison, WI, USA)[41]. Non-parenchymal cell fraction was resuspended in a solution with 9 ml ficoll and 1 ml percoll. One milliliter PBS was added carefully and spin at 4 °C for 15 min at $1400 \times g$ in centrifuge. HSC cell fraction was obtained from the layer between PBS and the ficoll/percoll gradient solution. Kupffer cells were isolated using Optiprep[38]. The nonparenchymal cells are resuspended in 20% Optiprep. HBSS and 11.5% Optiprep were layered on the cell suspension and centrifuged at $1811 \times g$ for 17 m. Kupffer cell fraction was obtained from the layer between 20% Optiprep and 11.5% Optiprep.

**Table 1 Antibodies.**

| Target | Supplier | Cat.no | Working dilutions 1st, 2nd antibody |
|---|---|---|---|
| αSMA | Sigma | A5228 | 1:1000, 1:2000 |
| NOS2 | Santa cruz | sc-650 | 1:1000,1:2000 |
| COX2 | Cell Signaling | 4842S | 1:1000,1:1000 |
| GAPDH | Millipore | CB1001 | 1:5000, 1:5000 |
| NLRP3 | Adipogen | AG-20B-0014-C100 | 1:1000, 1:2000 |
| IL-1β | Abcam | 9722 | 1:500, 1:2000 |
| Caspase-1 | Santa cruz | sc-56036 | 1:2000, 1:4000 |
| Caspase-1, cleaved | Adipogen | AG-20B-0042-C100 | 1:500, 1:1000 |
| ASC | Adipogen | AG-25B-0006-C100 | 1:1000, 1:2000 |
| GSDMDC1 | Santa cruz | sc-393656 | 1:500, 1:1000 |
| HMGB1 | Cell Signaling | 3935S | 1:500, 1:1000 |
| Caspase-3 | Cell Signaling | 9662S | 1:2000, 1:4000 |
| Caspase-3, Cleaved | Cell Signaling | 9664S | 1:500, 1:1000 |
| psmad3 | Cell Signaling | 9520S | 1:1000, 1:2000 |
| smad3 | Cell Signaling | 9523S | 1:1000, 1:2000 |
| psmad2 | Cell Signaling | 3108S | 1:1000, 1:2000 |
| smad2 | Cell Signaling | 5339S | 1:1000, 1:2000 |
| TrxR2 | Santa cruz | sc-376868 | 1:500, 1:2000 |
| Trx2 | Santa cruz | sc-50336 | 1:500, 1:2000 |
| TrxR1 | Santa cruz | sc-28321 | 1:500, 1:2000 |
| Trx1 | Cell Signaling | 6925S | 1:1000,1:2000 |
| xCT | Novus | NB300-318 | 1:500, 1:2000 |
| CD44 | Abcam | ab51037 | 1:1000, 1:2000 |
| CBS | Santa cruz | sc-133154 | 1:1000, 1:4000 |
| CTH | Abnova | H00001491-M01 | 1:1000, 1:4000 |

**Western blot analysis**. Liver tissues were lysed with RIPA lysis buffer. For assessment of secreted IL-1β and active caspase-1, cell culture medium was collected and precipitated using trichloroacetic acid (TCA, Sigma, St. Louis, MO, USA) and centrifuged at $20,000 \times g$ for 15 min. The sedimented proteins were dissolved in 1x SDS sample buffer. Cellular proteins were lysed with Triton X-100 lysis buffer (10 mM Tris, 100 mM NaCl, 1 mM EGTA, 10% Glycerol, 1% TritonX-100 and 30 mM sodium pyrophosphate). Prepared samples were separated using sodium dodecyl sulfate-polyacrylamide gel electrophoresis (SDS–PAGE). The fractionated proteins were transferred to nitrocellulose paper (GE healthcare, Madison, WI, USA). Then membranes were blocked with 5% skim milk and then incubated with the primary antibodies (Main Table 1). Horseradish peroxidase conjugated IgG antibodies (Cell Signaling Technology, Beverly, MA, USA) were used as the secondary antibodies. Immune complexes were detected by using the Immobilon Western Chemiluminescent HRP Substrate (Merck Millipore, Billerica, MA, USA), Densitometric protein levels were quantified by the image software, Multi Gauge, V3.0 (FUJIFILM, Tokyo, Japan).

**RNA preparation and real-time quantitative polymerase chain reaction (qPCR)**. RNA was isolated using Trizol® reagent (Thermo Fisher Scientific, Waltham, MA, USA) and Oligo (dT) primers (Main Table 2) and reverse transcriptase (iNtRON Biotechnology, Seongnam, Korea) were used to synthesize cDNA. mRNA expression levels of fibrosis marker, were quantified by real-time qPCR using a SYBR Select Master Mix (Applied Biosystems, Foster City, CA, USA) and Bio-Rad CFX Manager™ Software (Bio-Rad, Hercules, CA, USA).

**Table 2 Oligo sequences.**

| Name | Sequence | Supplier |
|---|---|---|
| *ACTA2* primer for qPCR | Forward(5′→3′) CTTTGACTTGCCGCCTACAC Reverse(5′→3′) ACAGTAAAGCCTGACCCCAA | Bioneer |
| *COL1A1* primer for qPCR | Forward(5′→3′) ATCTCCTGGTGCTGATGG Reverse (5′→3′) GCCTCTTTCTCCTCTCTGA | Bioneer |
| *TGFB1* primer for qPCR | Forward(5′→3′) GGATTTTGCCCCTTCGTTCC Reverse(5′→3′) GCCCATTTCCTGGTCGTGTT | Bioneer |
| *YM1* primer for qPCR | Forward(5′→3′) CATGAGCAAGACTTGCGTGAC Reverse(5′→3′) GGTCCAAACTTCCATCCTCCA | Bioneer |
| *ARG1* primer for qPCR | Forward(5′→3′) GGAAAGCCAATGAAGAGCTG Reverse(5′→3′) GCTTCCAACTGCCAGACTGT | Bioneer |
| CIITA primer for qPCR | Forward(5′→3′) AAGAGAAGGCTGGAAGGATCTTT Reverse(5′→3′) GATGTGGAAGACCTGGATCGT | Bioneer |
| CD44 primer for qPCR | Forward(5′→3′) ACAGTACCTTACCCACCATG Reverse(5′→3′) GGATGAATCCTCGGAATT | Bioneer |
| *SLC7A11* primer #1 for conventional PCR | Forward(5′→3′) AGAATTATGAACTTAATGCA Reverse(5′→3′) CCCAGTAGGTAAAGCTATGTT | Bioneer |
| *SLC7A11* primer #2 for conventional PCR and qPCR | Forward(5′→3′) CCTGGCATTTGGACGCTACAT Reverse(5′→3′) TCAGAATTGCTGTGAGCTTGC | Bioneer |

**Enzyme-linked immunosorbent assay (ELISA).** Culture media were collected and centrifuged to pellet down non-adherent cells as well as debris. Commercial ELISA kits (mouse TNFα and IL-1β, both from Invitrogen, Carlsbad, CA, USA) were used to measure cytokine concentration following the manufacture's protocol.

**Measurement of cell viability.** 3-(4,5-dimethylthiazol-2-yl)−2,5-diphenyl tetrazolium bromide (MTT, Sigma, St. Louis, MO, USA) assay was used to assess half-maximal inhibitory concentration (IC50). In all, 0.3 mg/ml MTT was added to each well of 48-well plates and the cells were incubated for 2 h. The culture media were removed and the resident cells were dissolved in 200 μL dimethylsulfoxide. The absorbance was measured at 590 nm with Tirstar microplate reader (Berthold Technologies, Bad Wildbad, Germany).

**IncuCyte® caspase-3/7 green apoptosis assay.** For detection of apoptosis, IncuCyte® caspase-3/7 green apoptosis assay was performed. Incucyte® caspase-3/7 green reagent (Essen BioScience, Ann Arbor, MI, USA) was diluted to culture medium to a final concentration of 1 μM. Using IncuCyte® live-cell analysis system, fluorescent objects were quantified every 4 h after adding the reagent to culture medium.

**Chemotaxis assay.** Activated HSCs and LX-2 cells were plated in the upper chambers of Incucyte® clearview 96-well cell migration plate (Essen BioScience, Ann Arbor, MI, USA). In order to facilitate cell movement, 10% FBS was added to reservoir plate as a chemoattractant. Migrated cells were counted using IncuCyte® live-cell analysis system. For transwell migration assay, LX-2 cells were plated in the upper chambers of type 1 collagen (Sigma, St. Louis, MO, USA) coated transwell polycarbonate membrane culture inserts (Corning, New York, USA). Medium supplemented with 10% FBS was added to the lower chamber. After 24 h culture, LX-2 cells moved to lower surface of the chamber were stained with 0.1% crystal violet and counted.

**Quantification of intracellular auranofin by inductively coupled plasma-mass spectrometry (ICP-MS).** BMDMs were seeded in 60 mm dish in a density of $3 \times 10^6$ cells per dish and treated with auranofin (1 or 10 μM) for 1 h. Supernatants were collected and BMDMs were harvested after washing with DL-dithiothreitol (Sigma, St. Louis, MO, USA) containing PBS for three times. Gold element in the samples was measured using inductively coupled plasma-mass spectrometer (ICP-MS), Varian 820-MS (Varian Australia Pty Ltd, Victoria, Australia).

**GSH assay.** Total intracellular glutathione GSH (GSH + GSSG) concentrations were determined using the GSH assay kit (Cayman Chemical, Ann Arbor, MI, USA) following the manufacturer's instruction. BMDMs were collected for analysis of intracellular GSH and sonicated.

**Human tissue sample and immunofluorescence staining.** Archived formalin-fixed paraffin-embedded (FFPE) tissue and frozen tissue from liver biopsy samples, which were obtained from non-alcoholic fatty liver disease patients and confirmed by pathologist, were used. The control normal liver tissue was obtained from patients through hepatic resection for extended cholecystectomy or blunt trauma. All control subjects had normal liver enzymes and <5% hepatic fat content. This study was approved by Institutional Review Board (IRB) of Hanyang University Hospital (HYUH 2019-12-028-006). FFPE tissue sections were deparaffinized and rehydrated. Heat-induced antigen retrieval was performed for 20 min in sodium citrate buffer (pH 6.0).

Sections were incubated with anti-αSMA antibody (dilution, 1:100) and anti-xCT antibody (dilution, 1:100) for 30 min at room temperature. The tissue sections were then incubated with secondary antibodies conjugated with Alexa Fluor 488 and 594 for 30 min at room temperature. The nuclei were stained with DAPI. The sections were examined using immunofluorescence microscope, and images were acquired with digital camera attached on the microscope.

**Cystine uptake assay using stable isotope-labeled cystine.** U-$^{13}C_3$-cysteine was oxidized to U-$^{13}C_6$-cystine using 30% hydrogen peroxide according to the previous methods[42,43]. Thirty percent hydrogen peroxide solution was mixed with 5-fold volume of 0.56 M U-$^{13}C_3$-cysteine in sterile distilled water. The purity of U-$^{13}C_6$-cystine was measured as 99.5% by HPLC, and cysteine, cysteine sulfinic acid and cysteic acid were not detected. The harvested cell lysate samples were diluted as 3-fold with 80% methanol containing 2 μM $^{13}C_5$, $D_5$, $^{15}N$-glutamate as an internal standard and reduced with 0.3 mM Tris-(2-carboxyethyl)-phosphine hydrochloride at room temperature for 30 min to analyze total $^{13}C_3$-cysteine. The samples were centrifuged at $10,000 \times g$, 4 °C for 10 min and the supernatant was injected to Applied Biosystems SCIEX 4000 QTRAP hybrid triple quadrupole-linear ion trap mass spectrometer equipped with Agilent 1290 Infinity II system. Acquisition and data analysis were performed with Analyst® software (ver.1.6.2; Applied Biosystems).

**Statistics and reproducibility.** Data are mean ± SD. Statistical significance between two groups was evaluated using the student's *t*-test, while comparisons of multiple groups were assessed by one-way ANOVA, followed by the Tukey's test. *p*-value < 0.05 was considered statistically significant. GraphPad PRISM software (GraphPad Software, Inc.) was used for all statistical analysis.

**Reporting summary.** Further information on research design is available in the Nature Research Reporting Summary linked to this article.

## Data availability

RNA-seq data have been deposited in the NCBI Gene Expression Omnibus under accession number GSE176346. Full gels/blots with molecular markers are provided in the Supplementary information. Data sets analyzed in this study are included in the Supplementary Data 1.

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

## Acknowledgements

This work was supported by the National Research Foundation of Korea (NRF) grants funded by the Korean Government (2017M3A9C8028794 and 2021R1A4A1021787). The first author, Hyun Young Kim was supported by the Korea Research Foundation (Scholarship program 2019R1A6A3A13096191).

## Author contributions

K.W.K. contributed concept and design of the study, writing article. H.Y.K. contributed concept and design of the study, experiments, procedures and writing article. S.C.L. contributed concept and design of the study, experiments and procedures. Y.J.C., H.K., K.Y., and N.K. contributed experiments and procedures. S.K.K., D.W.J., J.W.H., and Y.K. contributed concept and design of the study.

## Competing interests
The authors declare no competing interests.
