## [Peer Review File · Communications Biology]

Reviewers' Comments:

Reviewer #1:

Remarks to the Author:

Here the authors described the effect of auranofin on NLRP3 inflammasome activation and its anti-fibrotic effects. They also identified the system Xc as a potential mechanism, although the data is not sufficient to affirm Xc is necessary for the auranofin effects. One of my major concern is that the majority of the data are based on in vitro experiments and on BMDM. Are these effects observed in resident macrophages? The authors also did not prove that GSH depletion is either sufficient or necessary for the effects of auranofin on inflammasome. Inflammasome activation is investigated by using LPS plus ATP, but this is not what is going on in vivo in liver. What are the priming agents and what is triggering inflammasome activation in vivo, are these affected by auranofin? What happens upon in vivo deletion of NLRP3? Is the mechanism really NLRP3-dependent?

It is well demonstrated that in macrophages RET-derived ROS stabilizes HIF-1alpha to induce metabolic reprogramming and IL-1beta expression. The authors show that NAC treatment reversed the inhibitory effects of auranofin on IL-1beta secretion. Furthermore, it is well described that treatment of LPS-activated macrophages with NAC impairs IL-1beta expression, which contrasts with the data showed in this manuscript. Another point is the effects of Xc on NLRP3 inflammasome, can these effects be observed in vivo? Are they specific for NLRP3 inflammasome? Does it affect IL-1beta expression (priming)?

Minor concerns:

- 1 - The authors show reduced liver fibrosis, but no change in collagen deposition.
- 2 - There is no cleaved caspase in Fig 2b as mentioned. It also appears that there is no difference in cleaved IL-1beta.
- 3 - Does auranofin also modify IL-1beta expression?
- 4 - Sup Fig 2a show arginase and GAPDH and no data on AIM2 or NLRC4.
- 5 - Only morphology is not sufficient to determine pyroptosis.
- 6 - Data in Figure 3 is not sufficient to determine that auranofin changes the "death pathway".
- 7 - Replace death-associated molecular patterns for damage-associated molecular patterns.
- 8 - The WB in Fig 1e looks strange, cannot distinguish between samples.
- 9 - WB in Fig 2b seems different. It seems that the load control is from another blow/run. Please, included the full blots as supplementary material, at least for revision.

Reviewer #2:

None

Reviewer #3:

Remarks to the Author:

The manuscript entitled 'Auranofin prevents liver fibrosis by system Xc mediated inhibition of NLRP3 inflammasome' authored by Kim et al. describes that Auranofin inhibited system Xc activity and instantly induced oxidative burst, which mediated inhibition of the NLRP3 inflammasome in macrophages and HSCs. It is an intrested story. However, in vivo experiments presented in the manuscript are not convincing in support of the claimed conclusions due to concerns about the

experimental design in the studies.

1. Animal experiment design

Mice should be randomly divided into eight groups (Vehicle, TAA, Vehicle+AF 1 mg/Kg, TAA+AF 1 mg/Kg, Vehicle+AF 3 mg/Kg, TAA+AF 3 mg/Kg, Vehicle+AF 10 mg/Kg, TAA+AF 10 mg/Kg). The authors miss out Vehicle+AF 1 mg/Kg, Vehicle+AF 3 mg/Kg and Vehicle+AF 10 mg/Kg groups.

2. In Figure 6e and supplementary Figure 3c, The authors lack the condition of 'LPS+ATP'.

3. In Figure 8e, f, 'Sulfasalazine', 'Erastin' are labeled in wrong places.

Reviewer #4:

Remarks to the Author:

Liver diseases have a widespread epidemic in the world, which affect human health and quality of life. The causes of liver diseases are complicated and vary among different individuals. Inflammatory response, apoptosis, and fibrosis are important common mechanisms contributing to liver diseases. Studies have shown that inhibiting liver inflammation reduces liver damage and liver fibrosis. The authors found that a clinical anti-rheumatic agent, auranofin, can cure liver fibrosis by inhibiting the activation of NLRP3 in mouse models of liver fibrosis and various cell experiments. Totally, the authors proposed that auranofin could affect the ROS levels of macrophages and HSCs by regulating the activity of xCT to change the intracellular cysteine concentration, thereby inhibiting NLRP3 activation and IL-1 β release. This study puts forward new views on the functions and mechanisms of a clinical drug, and showed potential value for clinical treatment of liver fibrosis. However, there are still a few errors and flaws in the manuscript. Specific comments are listed below:

1 The author believes that xCT is up-regulated in the liver of NASH model mice, but the corresponding immunofluorescence data (Fig. 8 c) is of low quality and the expression level needs to be further confirmed by WB.

2 The immunohistochemical data in Fig. 8 d cannot clearly determine the positive signal. The staining quality must be improved and a partially enlarged photo must be provided.

3 Regarding the controversy about the regulation of inflammatory response by ROS, this article believes that short-term ROS bursts inhibit NLRP3 activation, so how will moderate levels of ROS work in this research model?

4 The results described in lines 160-162 in the main text correspond to incorrect Figs.

Reviewer #1 (Remarks to the Author):

Here the authors described the effect of auranofin on NLRP3 inflammasome activation and its anti-fibrotic effects. They also identified the system Xc as a potential mechanism, although the data is not sufficient to affirm Xc is necessary for the auranofin effects. One of my major concern is that the majority of the data are based on in vitro experiments and on BMDM. Are these effects observed in resident macrophages?

- Although kupffer cells initiate inflammation in the liver, hepatic damage results in the massive infiltration of bone marrow monocyte-derived macrophages (1). In the revised manuscript, we demonstrated that the effects of auranofin on NLRP3 inflammasome in kupffer cells were consistent with those in BMDM. IL-1 β secretion in kupffer cells by LPS/ATP or LPS/nigericin was mitigated by treatment of auranofin. (Fig. 2g in the revised manuscript).

Furthermore, the effects of xCT inhibition on NLRP3 inflammasome were confirmed in kupffer cells. Specific xCT inhibitors, erastin (5, 10 μ M), and sulfasalazine (300 μ M) significantly decreased ATP-induced IL-1 β secretion from LPS-primed kupffer cells (Fig. 8h and 8i).

The authors also did not prove that GSH depletion is either sufficient or necessary for the effects of auranofin on inflammasome.

- In the revised manuscript, we provided evidence that GSH depletion causes NLRP3

inflammasome inhibition. We quantified intracellular GSH and IL-1 β secretion levels in BMDM treated with GSH-depleting reagents. Buthionine sulfoximine (BSO), a specific inhibitor of γ -glutamylcysteine-ligase depletes the intracellular GSH pool within 5 h (Supplementary Fig. 4b). While treatment of BSO for 1 h did not inhibit ATP-induced activation of the NLRP3 inflammasome, sustained exposure to BSO for 5 h significantly decreased intracellular GSH as well as inflammasome mediated IL-1 β secretion (Fig. 5c). We also used GSH conjugating reagent diethylmaleate (DEM) to further verify the correlation between GSH level and NLRP3 inflammasome. DEM treatment significantly reduced intracellular GSH within 1 h (Supplementary Fig. 4b). Inflammasome-mediated IL-1 β secretion level was significantly decreased by DEM at this time point (Fig. 5d). These results support a notion that GSH depletion leads to decreased IL-1 β secretion by auranofin.

Inflammasome activation is investigated by using LPS plus ATP, but this is not what is going on in vivo in liver. What are the priming agents and what is triggering inflammasome activation in vivo, are these affected by auranofin? What happens upon in vivo deletion of NLRP3? Is the mechanism really NLRP3-dependent?

- Modified LDL and circulating free fatty acids (FFAs) are endogenous mediators that have been found to activate the NLRP3 inflammasome (2). The liver plays a principal role in lipid metabolism by taking up FFA and storing lipid metabolites. Indeed, it is well established that FFAs including palmitic acid activate NLRP3 inflammasome (3, 4). In an attempt to evaluate the effect of auranofin on NLRP3 inflammasome activated by FFA, LPS-primed BMDMs were cotreated with palmitic acid (400 μ M) and auranofin for 20 h. Auranofin inhibited LPS/palmitic acid-induced IL-1 β secretion (Fig. 2e).

- According to recent studies, hepatic stellate cell-specific NLRP3-knockout, as well as systemic ablation of NLRP3, showed marked protection from liver fibrosis (5, 6). Moreover, the liver from NLRP3 knock-in mice exhibited severe inflammation and increased mRNA levels of profibrotic genes (7). These results demonstrate that NLRP3 is a key factor in the progression of liver fibrosis.

It is well demonstrated that in macrophages RET-derived ROS stabilizes HIF-1alpha to induce metabolic reprogramming and IL-1beta expression. The authors show that NAC treatment reversed the inhibitory effects of auranofin on IL-1beta secretion. Furthermore, it is well described that treatment of LPS-activated macrophages with NAC impairs IL-1beta expression, which contrasts with the data showed in this manuscript.

- LPS-primed macrophages were incubated with N-acetylcysteine (NAC) just for 1 h. Since less than 2 h is insufficient to modify pro-IL-1β protein expression (8), we excluded the possibility that NAC treatment affects the auranofin's effect on pro-IL-1β protein expression. In the revised manuscript, we found that the protein expression level of pro-IL-1β was not altered in LPS-primed macrophages 1 h after NAC treatment (Supplementary Fig. 4d). Moreover, it has been reported that NAC up to 20 mM at pH-adjusted medium does not affect activation signal of NLRP3 inflammasome (9).

Another point is the effects of Xc on NLRP3 inflammasome, can these effects be observed in vivo? Are they specific for NLRP3 inflammasome? Does it affect IL-1beta expression (priming)?

- Recently, Choi et al. have reported that pharmacologic inhibition of xCT attenuated alcoholic steatosis (10). The authors suggested a bidirectional interaction between hepatocytes and nonparenchymal cells as the driver of alcoholic steatosis. They used sulfasalazine, a well-known xCT inhibitor to prove the contribution of system Xc- on alcoholic steatosis. In our study, we further found evidence that the inhibition of system Xc- attenuates liver fibrosis via

blockade of NLRP3 inflammasome. Sulfasalazine and erastin were used to assess the effects of system Xc- on NLRP3 inflammasome. In both BMDMs and kupffer cells, treatment of sulfasalazine and erastin inhibited activation of the NLRP3 inflammasome.

To the reviewer's question of whether the effects of system Xc- on NLRP3 inflammasome can be observed *in vivo*, we propose system Xc- as a primary mechanism underlying decreased NLRP3 inflammasome signals in auranofin-administered mice. As shown in Figure 2b, administration of auranofin reduced the protein expression of NLRP3 inflammasome components (NLRP3, ASC and pro-IL-1 β) as well as inflammasome activation markers [mature IL-1 β and caspase-1 (p20)] in TAA-induced fibrotic liver tissues. The mechanism may not be specific for the activation signal of NLRP3 inflammasome because priming signals could be also affected by long-term exposure to auranofin (8 weeks). However, the activation signal of the NLRP inflammasome is a highly sensitive target for low concentration ranges of auranofin (~30 nM) in liver cells.

We further measured the pro-IL-1 β expression in BMDM incubated with xCT-specific inhibitor, erastin or sulfasalazine. Incubation of BMDM with xCT inhibitors for 1 h did not inhibit pro-IL-1 β expression. This demonstrates that xCT inhibitor does not affect priming signals of NLRP3 inflammasome when it is treated with ATP for 1 h (Supplementary Fig. 4e).

Minor concerns:

1 – The authors show reduced liver fibrosis, but no change in collagen deposition.

: We quantified collagen deposition using qPCR and western blot analyses. Administration of auranofin significantly decreased mRNA expression level of COL1A1 in TAA-induced fibrotic liver (Figure 1f). Collagen I protein level was decreased by administration of auranofin in CCl4-induced fibrotic liver (Supplementary Fig. 1c).

2 - There is no cleaved caspase in Fig 2b as mentioned. It also appears that there is no difference in cleaved IL-1beta.

: We included caspase-1 (p20) blot in the revised Figure 2b. Moreover, we quantified the expression of mature IL-1 β using densitometric analyses and verified a decrease in the liver tissues from the auranofin administered group. A mature form of IL-1 β is hard to detect in *in vivo* samples because most of the mature IL-1 β is secreted to serum after its cleavage.

3 - Does auranofin also modify IL-1beta expression?

: Auranofin treatment for 4 h exerts an anti-inflammatory effect by repressing the mRNA expression of the NLRP3/IL-1 β pathway (11). In this study, however, IL-1 β secretion was measured after treatment of auranofin to LPS primed BMDM for 1 h. Because less than 2 h is too short to modify pro-IL-1 β protein translation, we did not consider the effect of auranofin on pro-IL-1 β expression.

4 - Sup Fig 2a show arginase and GAPDH and no data on AIM2 or NLRC4.

: We added Sup Fig 2a to demonstrate the effect of auranofin on the differentiation of M2 macrophages. Expression arginase-1 is a representative marker of M2 macrophage. We measured the mRNA expression of additional M2 markers (YM1, CIITA) and included the data in the revised Supplementary Fig. 2a.

5 – Only morphology is not sufficient to determine pyroptosis. / 6 - Data in Figure 3 is not sufficient to

determine that auranofin changes the “death pathway”.

: The images of cellular morphology only indicate that the auranofin-induced cell death pathway is different from that of LPS/ATP treated BMDM. As shown in Figure 3, we found that auranofin treatment induces apoptosis even when BMDM is in a condition of pyroptosis. Pyroptosis is a programmed, inflammatory cell death that is dependent on the activation of caspase-1. Cleavage of gasdermin D (GSDMD) is an executor of pyroptosis. N-terminal fragment of GSDMD (GSDMD-N) generated by caspase-1 forms membrane pores and releases intracellular contents, including high mobility group box 1 (HMGB1) (12). 16 h ATP treatment to LPS primed BMDMs induced caspase-1 (p20) and GSDMD-N. Cotreatment of auranofin with ATP, however, induced cleavage of caspase-3, an executor of apoptosis.

7 - Replace death-associated molecular patterns for damage-associated molecular patterns.

: We corrected it in the revised manuscript.

8 - The WB in Fig 1e looks strange, cannot distinguish between samples.

: Reflecting on your comment, we changed the α SMA blot. To enhance the quality of the blots, we optimized loading concentrations and used another α SMA antibody (Sigma, A5228). Revised western blots are listed below. We quantified α SMA expressions in thioacetamide (TAA) injected mice (n = 7-8 mice per group) using α SMA expressions using densitometric analyses. We used the first blot as a representative sample and inserted its figure in the revised manuscript.

9 - WB in Fig 2b seems different. It seems that the load control is from another blow/run. Please, included the full blots as supplementary material, at least for revision.

: Full blots were included for revision.

1. Tacke F, Zimmermann HW. Macrophage heterogeneity in liver injury and fibrosis. *Journal of hepatology* 2014;60:1090-1096.
2. De Nardo D, Latz E. NLRP3 inflammasomes link inflammation and metabolic disease. *Trends in immunology* 2011;32:373-379.
3. Wen H, Gris D, Lei Y, Jha S, Zhang L, Huang MT-H, Brickey WJ, et al. Fatty acid-induced NLRP3-ASC inflammasome activation interferes with insulin signaling. *Nature immunology* 2011;12:408-415.
4. Dong Z, Zhuang Q, Ning M, Wu S, Lu L, Wan X. Palmitic acid stimulates NLRP3 inflammasome activation through TLR4-NF- κ B signal pathway in hepatic stellate cells. *Annals of Translational Medicine* 2020;8:168.
5. Inzaugarat ME, Johnson CD, Holtmann TM, McGeough MD, Trautwein C, Papouchado BG, Schwabe R, et al. NLR family pyrin domain-containing 3 Inflammasome activation in hepatic stellate cells induces liver fibrosis in mice. *Hepatology* 2019;69:845-859.
6. Wree A, McGeough MD, Peña CA, Schlattjan M, Li H, Inzaugarat ME, Messer K, et al. NLRP3 inflammasome activation is required for fibrosis development in NAFLD. *Journal of molecular medicine* 2014;92:1069-1082.
7. Wree A, McGeough MD, Inzaugarat ME, Eguchi A, Schuster S, Johnson CD, Peña CA, et al. NLRP3 inflammasome driven liver injury and fibrosis: Roles of IL-17 and TNF in mice. *Hepatology* 2018;67:736-749.

8. Turner M, Chantry D, Buchan G, Barrett K, Feldmann M. Regulation of expression of human IL-1 alpha and IL-1 beta genes. *The Journal of Immunology* 1989;143:3556-3561.
9. Deigendesch N, Zychlinsky A, Meissner F. Copper Regulates the Canonical NLRP3 Inflammasome. *Journal of immunology (Baltimore, Md.: 1950)* 2018;200:1607.
10. Choi W-M, Kim H-H, Kim M-H, Cinar R, Yi H-S, Eun HS, Kim S-H, et al. Glutamate signaling in hepatic stellate cells drives alcoholic steatosis. *Cell metabolism* 2019;30:877-889. e877.
11. Isakov E, Weisman-Shomer P, Benhar MJBEBB-GS. Suppression of the pro-inflammatory NLRP3/interleukin-1 β pathway in macrophages by the thioredoxin reductase inhibitor auranofin. 2014;1840:3153-3161.
12. Liu X, Zhang Z, Ruan J, Pan Y, Magupalli VG, Wu H, Lieberman J. Inflammasome-activated gasdermin D causes pyroptosis by forming membrane pores. *Nature* 2016;535:153-158.

Reviewer #2 (Remarks to the Author):

The manuscript entitled 'Auranofin prevents liver fibrosis by system Xc mediated inhibition of NLRP3 inflammasome' authored by Kim et al. describes that Auranofin inhibited system Xc activity and instantly induced oxidative burst, which mediated inhibition of the NLRP3 inflammasome in macrophages and HSCs. It is an interesting story. However, in vivo experiments presented in the manuscript are not convincing in support of the claimed conclusions due to concerns about the experimental design in the studies.

1. Animal experiment design

Mice should randomly divided into eight groups (Vehicle, TAA, Vehicle+AF 1 mg/Kg, TAA+AF 1 mg/Kg, Vehicle+AF 3 mg/Kg, TAA+AF 3 mg/Kg, Vehicle+AF 10 mg/Kg, TAA+AF 10 mg/Kg). The authors miss out Vehicle+AF 1 mg/Kg, Vehicle+AF 3 mg/Kg and Vehicle+AF 10 mg/Kg groups.

: In the initial draft, we did not include the data of auranofin-only treated group because auranofin as a FDA approved drug showed no abnormal pathological changes in liver tissue samples. In fact, mice were divided into six groups (Vehicle, TAA, TAA+AF 1 mg/kg, TAA+AF 3 mg/kg, TAA+AF 10 mg/kg, AF 10 mg/kg). In the revised manuscript, liver tissue staining data and pathological analysis data of mice administered with AF 10 mg/kg alone (Vehicle+AF 10 mg/kg group, n=8) were presented as follows. High dose of auranofin (10 mg/kg) administration did not induce any significant hepatic damages (Figs. 1a-1d).

2. In Figure 6e and supplementary Figure 3c, The authors lack the condition of 'LPS+ATP'.

We appreciate the reviewer's careful indication. There was a mistake in the notation of the data. The value of ATP only treated group should be changed to the value of LPS+ATP treated group. We reflected this in the revised manuscript (Fig. 6e and supplementary Fig. 3c).

3. In Figure 8e, f, 'Sulfasalazine', 'Erastin' are labeled in wrong places.

We corrected the labels in the revised manuscript (Figs 8f and 8g).

Reviewer #3 (Remarks to the Author):

Liver diseases have a widespread epidemic in the world, which affect human health and quality of life. The causes of liver diseases are complicated and vary among different individuals. Inflammatory response, apoptosis, and fibrosis are important common mechanisms contributing to liver diseases. Studies have shown that inhibiting liver inflammation reduces liver damage and liver fibrosis. The authors found that a clinical anti-rheumatic agent, auranofin, can cure liver fibrosis by inhibiting the activation of NLRP3 in mouse models of liver fibrosis and various cell experiments. Totally, the authors proposed that auranofin could affect the ROS levels of macrophages and HSCs by regulating the activity of xCT to change the intracellular cysteine concentration, thereby inhibiting NLRP3 activation and IL-1 β release. This study puts forward new views on the functions and mechanisms of a clinical drug, and showed potential value for clinical treatment of liver fibrosis. However, there are still a few errors and flaws in the manuscript. Specific comments are listed below:

1. The author believes that xCT is up-regulated in the liver of NASH model mice, but the corresponding immunofluorescence data (Fig. 8 c) is of low quality and the expression level needs to be further confirmed by WB.

- Immunofluorescence data (Fig. 8c) was detected in liver samples from the NASH patients with fibrosis. Colocalization of xCT and α SMA appeared as yellow dots. Since the specificity of xCT antibody is batch-dependent in western blot experiments (1), we confirmed the xCT expression level using real-time quantitative PCR analysis. While SLC7A11 (xCT) was not detected (Cq value >35) in liver samples from normal mice, TAA-induced fibrotic liver samples expressed high levels of SLC7A11 (Fig. 8d).

2. The immunohistochemical data in Fig. 8 d cannot clearly determine the positive signal. The staining quality must be improved and a partially enlarged photo must be provided.

- Reflecting on the reviewer's suggestion, high magnification and high contrast images were used to clearly distinguish the positive signal (Fig. 8e).

3. Regarding the controversy about the regulation of inflammatory response by ROS, this article believes that short-term ROS bursts inhibit NLRP3 activation, so how will moderate levels of ROS work in this research model?

- The mechanism by which ROS modulates NLRP3 activity remains elusive. However, a plausible explanation for the inhibitory effect of ROS bursts on NLRP3 activation is that superoxide directly decreases caspase-1 activity. In SOD1-deficient macrophages, higher superoxide levels decreased the cellular redox potential. Consequently, it inhibited caspase-1 by reversible oxidation and glutathionylation of the redox-sensitive cysteine residues (2). The effect of superoxide was independent of LPS priming, only inhibiting the cleavage of caspase-1.

In the auranofin-treated BMDM, we confirmed the increased ROS levels using fluorescent ROS indicator CM-H2DCFDA. Based on the previous report, the intracellular ROS generated by auranofin is expected to oxidize or glutathionylate the active cysteine site of caspase-1. Indeed, incubation of auranofin for 1 h did not reduce priming signals of the NLRP3 inflammasome. It only inhibited the cleavage of caspase-1 and maturation of IL-1 β .

The results described in lines 160-162 in the main text correspond to incorrect Figs.

- In the revised manuscript, figures are now correctly matched to the results in lines 170-172.

1. Van Liefferinge J, Bentea E, Demuyser T, Albertini G, Follin-Arbelet V, Holmseth S, Merckx E, et al. Comparative analysis of antibodies to xCT (Slc7a11): forewarned is forearmed. *Nature immunology* 2016;17:1015-1032.
2. Meissner F, Molawi K, Zychlinsky A. Superoxide dismutase 1 regulates caspase-1 and endotoxic shock. *Nature immunology* 2008;9:866.

Reviewers' Comments:

Reviewer #1:

Remarks to the Author:

I am satisfied with the responses.

Reviewer #3:

None

Reviewer #4:

None

Reviewer #3 (Remarks to the Author):

The immunofluorescence data of xCT (Fig. 8c) may not be specific. xCT is a membrane-localized protein, but the results showed in the manuscript xCT does not appear to be localized to the cell membrane. The author believes that the antibodies of xCT used in this study were not suitable for WB testing, but the IHC data was provided in the manuscript. The specificity of xCT's IHC results is still uncertain. So it is not certain that xCT is up-regulated in fibrotic liver samples.

- The immunofluorescence data of the initial draft were obtained from formalin-fixed paraffin-embedded (FFPE) liver tissues from profibrotic NASH patients. As the reviewer's comment, we found out that the immunofluorescence images highly exhibited background autofluorescent signals in FFPE tissues. To minimize the effect of autofluorescence, we used a frozen liver section from a profibrotic NASH patient to confirm the location of xCT protein (Fig. 8c in the revised manuscript). xCT did appear to be mainly localized to the cell membrane.

Unavoidably, there were non-specific bands in western blots using xCT antibody (Supplementary Fig. 5a). However, IHC staining demonstrated that the expression of xCT was increased in liver tissues from NASH patients and TAA-induced fibrotic mouse liver tissues compared to those from the control group (Fig. 8a, b, e). Moreover, the analysis using the real-time qPCR additionally confirmed the upregulation of xCT in fibrotic liver samples from mice (Fig. 8d).

C